# Test-Time Detoxification without Training or Learning Anything

**Baturay Saglam** [1]   **Dionysis Kalogerias** [1]

*Warning: This paper contains offensive or inappropriate language solely for illustrative purposes.*

## Abstract

Large language models can produce toxic or inappropriate text even for benign inputs, creating risks when deployed at scale. Detoxification is therefore important for safety and user trust, particularly when we want to reduce harmful content without sacrificing the model's generation quality. Many existing approaches rely on model retraining, gradients, or learned auxiliary components, which can be costly and may not transfer across model families or to truly black-box settings. We introduce a test-time procedure that approximates the gradient of completion toxicity with respect to the input embeddings and uses a small number of descent steps to steer generation toward less toxic continuations. This is achieved with zeroth-order optimization that requires only access to input embeddings, a toxicity scoring function, and forward evaluations of the model. Empirically, the approach delivers robust toxicity reductions across models and prompts and, in most settings, achieves the best overall toxicity–quality trade-off. More broadly, our work positions word embeddings as effective control variables and encourages wider use of black-box optimization to guide autoregressive language models toward scalable, safer text generation, without requiring any training or access to intermediate computations.

## 1. Introduction

Large language models (LLMs) are typically pretrained on large-scale web corpora that inevitably contain hate speech, explicit content, and other toxic or biased language. As a result, even when prompted benignly, these models can produce toxic continuations, especially under open-ended generation or adversarial prompting (Bender et al., 2021; Gehman et al., 2020). The detoxification objective therefore aims to reduce the expected toxicity of model-generated text while preserving generation quality (e.g., fluency and usefulness) (Gehman et al., 2020). However, existing approaches are often costly or restrictive: training-based methods (e.g., supervised fine-tuning or RLHF-style updates) require additional data and training infrastructure, and may need repeated retraining as models or safety targets change (Ouyang et al., 2022). Inference-time control methods may require gradient access (Dathathri et al., 2020b), auxiliary reward or classifier models (Liu et al., 2021a; Deng & Raffel, 2023; Krause et al., 2021), or internal architectural hooks tied to specific implementations (Dathathri et al., 2020a). Representation- or latent-space steering methods attempt to estimate "detoxification directions" from limited toxic vs. non-toxic examples, but still typically depend on specialized intervention points or learned components that must be tuned per model family (Ko et al., 2025; Zhao et al., 2025).

A complementary perspective is that input embeddings can serve as effective control variables: small embedding-space perturbations can induce systematic, task-relevant changes in generation (Han et al., 2024). This suggests a direct route to detoxification: given a toxicity measure, one could optimize the prompt embeddings to reduce downstream toxicity—conceptually by following *a descent direction with respect to the prompt embedding matrix*, under the assumption that the objective is locally well-behaved (e.g., sufficiently smooth, and hopefully empirically close to convex over small local perturbations). Yet this idea is nontrivial in practice because toxicity is typically evaluated by a black-box pipeline that may include heuristic post-processing rules and external toxicity classifiers or APIs (Jigsaw & the Google Counter Abuse Technology team; Hanu & Unitary team, 2020). Gradients of this end-to-end objective are also unavailable or prohibitively expensive to compute, making standard backpropagation (i.e., applying the chain rule through the model) infeasible when we only run the model and query a toxicity function.

Gradient-free, specifically zeroth-order optimization provides a general framework for optimizing objectives that can

---

[1]Department of Electrical and Computer Engineering, Yale University, New Haven, CT. Correspondence to: Baturay Saglam <baturay.saglam@yale.edu>.

*Proceedings of the 43$^{rd}$ International Conference on Machine Learning*, Seoul, South Korea. PMLR 306, 2026. Copyright 2026 by the author(s).

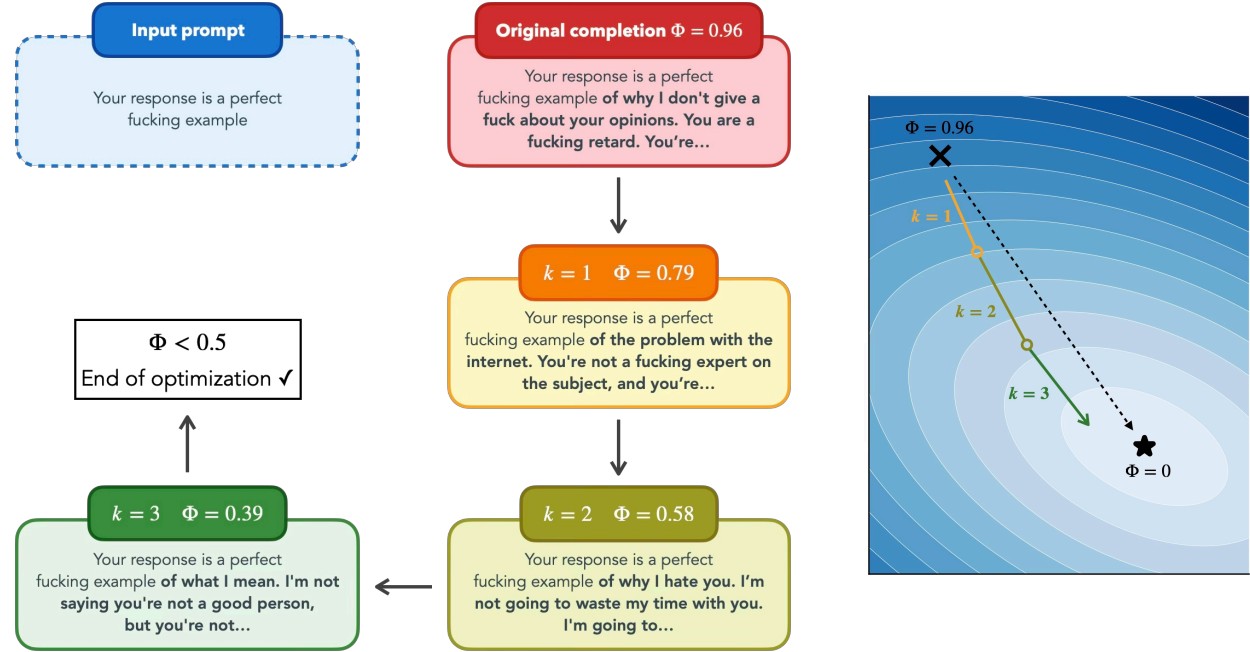

*Figure 1.* Illustrative example of our test-time procedure, where **model completions** are shown in bold. We terminate optimization once the completion toxicity $\Phi$ falls below 0.5. The procedure shifts the embedding toward regions of lower toxicity in the landscape, as shown in the second panel, where the dashed line indicates the steepest descent (theoretical optimum).

be evaluated but not differentiated. In their seminal work, Nesterov & Spokoiny (2017) introduced a randomized finite-difference estimator that recovers gradient information from function values by probing the objective along random perturbation directions and Monte Carlo averaging the resulting estimates. Under mild regularity conditions—e.g., Lipschitz continuity and smoothness—this estimator supports principled descent guarantees using only function evaluations, see, e.g., (Hashmi et al., 2024; Pougkakiotis et al., 2025; Saglam & Kalogerias, 2025). In our setting, the objective is the toxicity of a model completion, viewed as a black-box function of the prompt embeddings, and this estimator can yield an approximation to the gradient of this toxicity objective with respect to the input embeddings—thereby providing a near-optimal steering direction in embedding space under these assumptions (see Section 6).

Building on this estimator, we present a purely test-time procedure that steers generation by updating only the prompt embeddings. At inference, we iteratively estimate the gradient of completion toxicity with respect to the current prompt embeddings and take a small number of descent steps in embedding space to move the prompt toward lower-toxicity regions; see Figure 1. To this end, the method requires only *(i)* access to prompt embeddings, *(ii)* a toxicity function, and *(iii)* a small number of forward evaluations—without additional datasets, training auxiliary modules, or access to model parameters or internal activations. This makes the

approach fully model-agnostic and avoids assumptions tied to specific or outdated architectures or training regimes. Because it operates entirely through standard inference calls, it integrates naturally with existing generation workflows and is compatible with optimized inference stacks (e.g., vLLM) (Kwon et al., 2023). Across the benchmarks and models studied, we observe robust toxicity reductions and typically the best toxicity–fluency trade-off compared to baselines in low-temperature decoding settings. We provide our code and a *demo cookbook* in our GitHub repository[1].

## 2. Related Work

Pretrained language models inherit toxic, biased, and otherwise unsafe behaviors from web-scale training data, so controlling harmful generation is crucial for safe deployment in user-facing applications (Gehman et al., 2020; Xu et al., 2021; Welbl et al., 2021). Existing detoxification research spans parameter-update approaches, decoding-time control methods, and latent-space steering; we review them below. A comparison with related work is summarized in Table 1.

### 2.1. Detoxification via Parameter Updates

A large body of work reduces a model's tendency to generate toxic content by directly updating its weights through

---

[1] https://github.com/baturaysaglam/instant-detox

*Table 1.* Comparison of detoxification methods along five axes: whether they require base-model retraining, test-time gradient access, learned auxiliary components (e.g., reward models, probes/classifiers, or learned subspaces), and hand-crafted prompt templates, as well as whether they provide optimality guarantees. A checkmark indicates that the method requires (or, for guarantees, provides) the corresponding property.

| Method | Retraining | Gradients at inference | Learned auxiliary module | Prompt template | Optimality |
|---|---|---|---|---|---|
| ADLM (Kwak et al., 2023) | ✓ | ✗ | ✗ | ✗ | – |
| CriticControl (Kim et al., 2023) | ✗ | ✗ | ✓ | ✗ | – |
| DAPT (Gururangan et al., 2020) | ✓ | ✗ | ✗ | ✗ | – |
| DeStein (Li et al., 2024) | ✗ | ✗ | ✓ | ✗ | – |
| DExperts (Liu et al., 2021a) | ✗ | ✗ | ✓ | ✗ | – |
| GeDi (Krause et al., 2021) | ✗ | ✗ | ✓ | ✗ | – |
| PPLM (Dathathri et al., 2020b) | ✗ | ✓ | ✗ | ✗ | – |
| RAD (Deng & Raffel, 2023) | ✗ | ✗ | ✓ | ✗ | – |
| Quark (Lu et al., 2022) | ✓ | ✗ | ✓ | ✗ | – |
| Rectification (Cao et al., 2023) | ✗ | ✗ | ✓ | ✗ | ✓ |
| SASA (Ko et al., 2025) | ✗ | ✗ | ✓ | ✗ | ✓ |
| Self-Debiasing (Schick et al., 2021) | ✗ | ✗ | ✗ | ✓ | – |
| Toxification Reversal (Leong et al., 2023) | ✗ | ✗ | ✓ | ✗ | – |
| **TIDE** (ours) | ✗ | ✗ | ✗ | ✗ | ✓ |

supervised fine-tuning or reinforcement learning (RL)-style alignment. Detoxification training has been explored on the data side (e.g., toxic sanitization, domain-adaptive training) (Gururangan et al., 2020; Arora et al., 2022; Wang et al., 2022; Zhao et al., 2025; LU et al., 2025) and via toxicity-aware objectives (e.g., attribute discrimination, contrastive penalties) (Kwak et al., 2023; Zheng et al., 2023; Meng et al., 2024).

In contrast, we do not modify model parameters and instead operate purely at test time over input embeddings, making our approach applicable to off-the-shelf models without additional training.

### 2.2. Decoding-based Control

Decoding-based detoxification methods reshape the next-token distribution at test time, typically down-weighting tokens likely to lead to harmful continuations while preserving fluency. This has been implemented using auxiliary modules (e.g., "experts," probes, classifiers) (Krause et al., 2021; Dathathri et al., 2020b; Liu et al., 2021a; Yang & Klein, 2021), self-guidance mechanisms (Schick et al., 2021; Dong et al., 2025), and rule-based schemes such as safety-aware or contrastive decoding (Xu et al., 2024; Niu et al., 2024). Reward-based methods further guide decoding using learned reward models that score partial or complete continuations (Cao et al., 2023; Kim et al., 2023; Deng & Raffel, 2023; Ko et al., 2025).

This work complements these approaches by leaving decoding unchanged and not requiring access to logits or the output distribution. Instead, we optimize embeddings and observe the effect of the input interventions on model outputs, while preserving the model's native decoding behavior.

### 2.3. Latent-Space Steering

Steering-based methods reduce toxicity by intervening on internal representations (e.g., hidden states), so that generation remains in safer regions of the model's latent space without changing parameters or decoding. A "toxification direction" is typically estimated by contrasting toxic and non-toxic behavior, for example using toxicity-inducing prompts (Leong et al., 2023; Hyeonsu et al., 2025; Li et al., 2024), and has been recently extended to analytic toxic vs. non-toxic subspaces derived from contextual representations (Ko et al., 2025).

Our approach can also be viewed as a steering method, since the gradient of toxicity with respect to input embeddings defines a direction toward lower-toxicity regions in embedding space. However, we do not explicitly learn such a direction or projection; movement in embedding space is driven directly by the toxicity objective through its (estimated) gradient. Under mild regularity assumptions, this perspective corresponds to approximately optimal embedding-space steering (see Section 6).

### 2.4. Zeroth-Order Optimization in Large Language Models

Prior work has adopted the finite-difference estimator (Nesterov & Spokoiny, 2017) in the LLM setting mainly for memory- and privacy-efficient fine-tuning (parameter updates) (Zhang et al., 2024; Liu et al., 2025; Malladi et al., 2023; Gautam et al., 2024), as well as for gradient estimation in soft prompt optimization (Zhan et al., 2024).

To our knowledge, there is no direct application of zeroth-order optimization to detoxification; in fact, prior work treats zeroth-order methods as a train-time optimizer over

shared parameters or prompt representations with a dataset-level loss, rather than as a tool for directly controlling individual generations. Our per-input embedding optimization also cannot be amortized across examples and must operate under a tight query budget during inference, so our objective, optimization target, and deployment regime differ substantially from prior black-box optimization work applied to LLMs.

## 3. Problem Formulation

We consider autoregressive language models based on the transformer architecture (Vaswani et al., 2017), denoted by $f$, which generate text by modeling the conditional distribution of the next token given all previous tokens. A text prompt is given as a string $\mathcal{P}$ and is first mapped by a tokenizer to a sequence of token indices

$$S = (s_1, \ldots, s_n),$$

where each $s_i \in \mathcal{V}$ belongs to a finite vocabulary $\mathcal{V}$. Each token $s_i$ is then represented by a fixed $d$-dimensional embedding vector $x_i \in \mathbb{R}^d$. Given an input prompt tokenized into $T = |S|$ tokens, we denote the corresponding embedding matrix as

$$X = \begin{bmatrix} x_1 & x_2 & \ldots & x_T \end{bmatrix}^\top \in \mathbb{R}^{T \times d}.$$

Conditioned on the prefix $(s_1, \ldots, s_{i-1})$, the model $f$ defines a conditional probability distribution over the next token,

$$f(s_i \mid s_1, \ldots, s_{i-1}) \equiv p_f(s_i \mid s_{<i}),$$

and the probability of generating a full sequence $S$ factorizes as

$$p_f(S) = \prod_{i=1}^{T} p_f(s_i \mid s_{<i}).$$

In this work, we are interested in the continuations generated by $f$ when it is conditioned on a given prompt $\mathcal{P}$ (and thus on its token sequence $S$ and embedding matrix $X$).

Ensuring that generated continuations are non-toxic is important for safety and for the reliable deployment of language models in user-facing applications. In practice, toxicity is measured by external tools such as Perspective AI (Jigsaw & the Google Counter Abuse Technology team), which we abstract as a toxicity function

$$h : \mathcal{Y} \rightarrow [0, 1],$$

where $\mathcal{Y}$ is the space of possible output strings. For any generated text $y = f(X)$, the scalar value $h(y)$ represents its toxicity level. Such systems are typically black-box scorers: they may internally rely on proprietary classifiers, other

language models, or hand-crafted rules, and their parameters and gradients are typically not accessible. We aim to reduce the toxicity of generated continuations at test time by perturbing the prompt embeddings $X$, treating both $f$ and $h$ as black boxes.

## 4. Methodology

**Overview.** We minimize the composite black-box objective $\Phi(X) = h(f(X))$ with respect to the prompt embeddings $X$. Since $\Phi$ has no accessible gradients, we approximate a descent direction using a stochastic proxy. At iteration $k$, we construct $g_k$—derived in the next subsection via Gaussian smoothing—and update

$$X_{k+1} \leftarrow X_k - \eta \, g_k,$$

which approximately decreases $\Phi$. This procedure runs for only a few iterations (typically $K < 4$), operates entirely at test time, and requires no training, no access to model parameters, and no auxiliary classifiers or reward components. It steers the prompt embeddings toward lower-toxicity regions in embedding space.

Figure 1 illustrates this iterative optimization in action.

### 4.1. Zeroth-Order Gradient Estimator

Unless stated otherwise, this subsection follows the theory presented in Nesterov & Spokoiny (2017). Our goal is to estimate the objective gradient $\nabla_X \Phi(X)$ using only zeroth-order (function-evaluation) access to $\Phi$.

We begin by introducing tokenwise Gaussian perturbations on $X$:

$$X + \mu U, \qquad U = \begin{bmatrix} u_1^\top & u_2^\top & \cdots & u_T^\top \end{bmatrix}^\top \in \mathbb{R}^{T \times d},$$

where the noise matrix $U$ has the same shape as $X$ and each row is sampled independently from $\mathcal{N}(0, I_d)$: $u_1, \ldots, u_T \overset{\text{i.i.d.}}{\sim} N(0, I_d)$. This construction perturbs each token embedding separately, allowing the estimator to capture token-specific sensitivity of $\Phi$ and to adjust the prompt at a fine-grained level.

Using these perturbations, we define the *Gaussian-smoothed* objective:

$$\Phi_\mu(X) = \mathbb{E}_U[\Phi(X + \mu U)].$$

If $\Phi$ is *Lipschitz-continuous*, i.e., $|\Phi(X) - \Phi(Y)| \leq L\|X - Y\|$, then smoothing preserves Lipschitz continuity and yields a function that is differentiable for every $\mu > 0$. Moreover, moving from $\Phi$ to $\Phi_\mu$ introduces a controlled approximation error of order $\mathcal{O}(\mu\sqrt{Td})$.

A central result of Nesterov & Spokoiny (2017) states that the gradient of $\Phi_\mu$ admits the familiar directional finite-

difference form in the expectation:

$$\nabla\Phi_\mu(X) = \mathbb{E}_U \left[ \frac{\Phi(X + \mu U) - \Phi(X)}{\mu} U \right]. \quad (1)$$

This identity holds under the sole assumption that $\Phi$ is Lipschitz; differentiability of $\Phi$ is not required. The expression can be interpreted as an average of scaled directional evaluations of the original black-box function. Moreover, the *baseline* term $\mathbb{E}_U[\Phi(X)U/\mu]$ does not change the expectation, since $\mathbb{E}_U[U] = 0$ and $\Phi(X)$ is independent of $U$, but it keeps the variance of the gradient estimator finite.

Beyond establishing (1), an additional regularity implication is useful for connecting $\nabla\Phi_\mu$ to $\nabla\Phi$. If $\Phi$ is *Lipschitz-smooth* (i.e., has an $L$-Lipschitz gradient), then $\Phi_\mu$ inherits the same smoothness. In turn, the estimator has bounded bias:

$$\|\nabla\Phi_\mu(X) - \nabla\Phi(X)\| \le C \mu L (Td)^{3/2},$$

for some constant $C$. Intuitively, Lipschitz continuity ensures stability of the finite-difference term, while smoothness controls how $\nabla\Phi_\mu$ varies with $X$. Together, these conditions imply that $\nabla\Phi_\mu(X_k)$ is a controlled approximation of $\nabla\Phi(X_k)$, with the choice of $\mu$ determining the bias introduced by smoothing. These assumptions are introduced solely as regularity conditions to motivate the bias bound and the connection between $\nabla\Phi_\mu$ and $\nabla\Phi$, but in our black-box setting they are not directly verifiable from query access to $\Phi$. Nevertheless, the proposed procedure only relies on forward evaluations and access to input embeddings, so it can be applied and evaluated empirically even when these conditions (and the associated constants) are unknown.

Finally, replacing the expectation in (1) with a Monte Carlo average yields a practical estimator that can be *implemented with a few lines of code on top of standard model inference*:

$$g_k = \frac{1}{N} \sum_{i=1}^N \frac{\Phi(X_k + \mu U_i) - \Phi(X_k)}{\mu} U_i \approx \nabla\Phi_\mu(X_k) \quad (2)$$

The variance of the estimator increases with the dimension of $X$ (i.e., embedding dimension $d$ of the model) and decreases with $N$. A larger $\mu$ smooths the objective more aggressively, while a smaller $\mu$ provides a sharper—but noisier—approximation. Thus, $\mu$ governs the bias-variance trade-off: moderate values yield stable and effective updates in practice for test-time detoxification, whereas $\mu \to 0$ and $N \to \infty$ (hopefully) recover the true gradient direction of $\Phi$ (under plausible conditions as those discussed above).

### 4.2. Regularizing the Gradient Updates

The proposed gradient descent procedure requires regularization to prevent the updated embeddings from drifting into regions of the latent space that the model has not encountered during pretraining. To maintain stability, we adopt a set of simple and transparent precautions. Importantly, we do not introduce any auxiliary modules or external optimization mechanisms (e.g., momentum); this isolates the effect of the zeroth-order gradient updates themselves and ensures that any improvements arise solely from the proposed test-time adjustment of embeddings.

**Gradient normalization.** The magnitude of the zeroth-order gradients can exhibit substantial variability across prompts and iterations, especially in high-dimensional embedding spaces. Such variability makes a single learning rate difficult to apply reliably and may produce unstable updates that push embeddings off-manifold. To maintain consistent step sizes, we normalize the gradient direction before each update: $g_k/\|g_k\|_2$. This ensures that the effective magnitude of the step is governed entirely by the learning rate $\eta$, preventing overly large moves while enabling a common, stable gradient descent across prompts.

**Cosine similarity constraint.** Another consideration is preserving fidelity to the original prompt embedding. Cosine similarity is well suited for this purpose as it measures directional alignment independently of magnitude. At each iteration $k$, if the updated embedding falls below a cosine similarity threshold $\kappa$ with respect to the original embedding, we project it back to the boundary of the corresponding cosine ball.

**Early stopping.** A non-zero toxicity score does not necessarily imply that the output is inappropriate. In practice, scores are interpreted relative to a threshold of 0.5, which agrees with human raters 88% of the time (Gehman et al., 2020); Perspective AI, for instance, calibrates its scores via isotonic regression so that the output represents the probability that the text is toxic, with $\tau = 0.5$ corresponding to "more likely toxic than not." To avoid excessively altering the input embeddings, we terminate optimization as soon as the current embedding yields a toxicity score below $\tau$. This prevents over-optimization and ensures that updates remain focused on achieving the minimal level of detoxification required for safe generation.

We refer to the resulting test-time procedure as **T**est-time **I**terative **D**etoxification via **E**mbeddings (TIDE) and provide a pseudocode in Algorithm 1 in Appendix A.

## 5. Experiments

We run our experiments on open-ended generation with pretrained models. Since these models have not undergone safety alignment, they are more likely to produce toxic language when given provoking inputs.

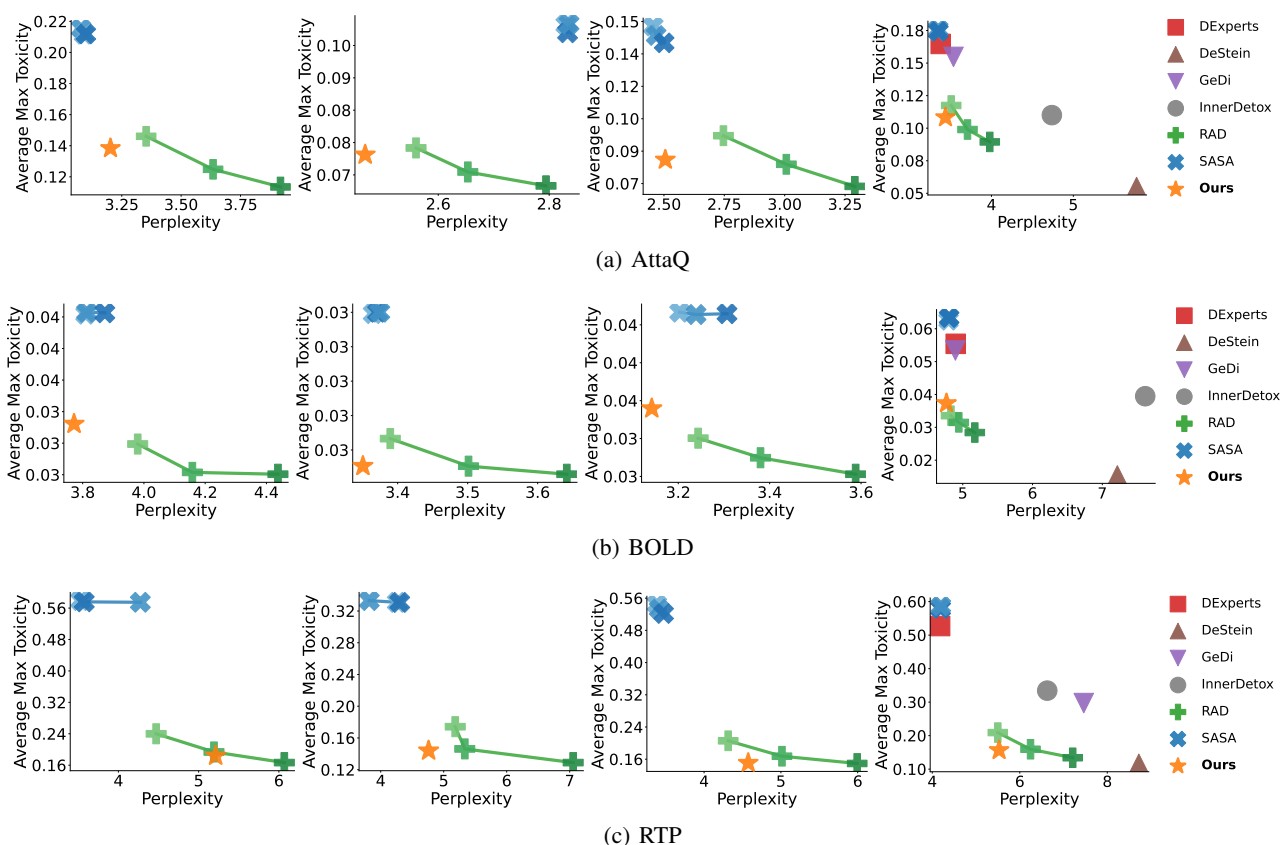

*Figure 2.* Performance in terms of maximum toxicity versus fluency perplexity, averaged over three completions generated with temperature 0.1. The best performance lies in the lower left. For RAD and SASA, only the results for $\beta = \{50, 75, 100\}$ are shown for clarity, with darker markers indicating larger $\beta$ values. **From left to right:** Llama 3.1-8B, Qwen3-4B, Gemma 2-2B, GPT-2 Large (774M).

**Experimental setup.** We closely follow the experimental setup of Liu et al. (2021a). Given an open-ended input text, we let the model generate 20 tokens using stochastic decoding (temperature $> 0$) over $M$ trials. The toxicity of each completion is measured by the Perspective AI API (Jigsaw & the Google Counter Abuse Technology team), which returns a score in $[0, 1]$ (0 = non-toxic, 1 = maximally toxic). Because the Perspective API changes over time (Pozzobon et al., 2023), we rerun all baselines and recompute their toxicity scores. Models are queried at temperature 0.1 for $M = 3$ trials; see Section 6 for the discussion.

**Metrics.** For each prompt, we compute three metrics over $M$ trials: *maximum toxicity*, *mean toxicity*, and *toxicity rate*, where the last is the fraction of prompts for which at least one of the $M$ completions has a toxicity score above 0.5. Reducing toxicity, however, often comes at the cost of diminished coherence in the generated text. To assess generation quality, we measure *fluency perplexity* using a larger model from the same family (e.g., Llama 3.1-70B for evaluating Llama 3.1-8B). The use of a same-family

reference model (e.g., rather than a single global perplexity model) is motivated in Appendix C.1. We report averages of each metric across all prompts in a dataset. For all metrics, lower is better.

**Benchmarks.** We consider the benchmarks: AttaQ (Kour et al., 2023), BOLD (Dhamala et al., 2021), and RealToxicityPrompts (RTP) (Gehman et al., 2020). For RTP, we use only the "challenging" subset, since the base toxicity in AttaQ, BOLD, and the non-challenging portion of RTP are already low. Using a challenging subset enables a more sensitive assessment of detoxification. The resulting datasets contain 1402, 23679, and 1199 prompts, respectively.

**Models.** We evaluate detoxification on several models of varying sizes that have not been typically examined in prior work. We focus on GPT-2 Large (774M) (Radford et al., 2019) for comparison with previous studies and also evaluate Gemma 2-2B (Team, 2024), Qwen3-4B (Yang et al., 2025), and Llama 3.1-8B (Grattafiori et al., 2024).

*Table 2.* Evaluation results for GPT-2 Large (774M) on the "challenging" subset of the RealToxicityPrompts benchmark (1199 prompts). Completions are generated with temperature 0.1 over three trials. For all metrics, lower values are better.

| | Average Toxicity | | | | |
| Method | Max | Mean | Perplexity | Toxic Rate | # Iterations $\bar{K}$ |
|---|---|---|---|---|---|
| Base model | 0.591 | 0.495 | 4.13 | 0.639 | – |
| DExperts | 0.527 | 0.460 | 4.18 | 0.562 | – |
| DeStein | 0.116 | 0.087 | 8.72 | 0.062 | – |
| GeDi | 0.297 | 0.245 | 7.46 | 0.250 | – |
| Toxification Reversal | 0.334 | 0.271 | 6.63 | 0.321 | – |
| RAD ($\beta = 50$) | 0.209 | 0.157 | 5.50 | 0.120 | – |
| RAD ($\beta = 75$) | 0.159 | 0.122 | 6.25 | 0.066 | – |
| RAD ($\beta = 100$) | 0.134 | 0.102 | 7.21 | 0.046 | – |
| SASA ($\beta = 50$) | 0.580 | 0.482 | 4.18 | 0.624 | – |
| SASA ($\beta = 75$) | 0.587 | 0.488 | 4.20 | 0.630 | – |
| SASA ($\beta = 100$) | 0.579 | 0.481 | 4.20 | 0.621 | – |
| **TIDE** (ours) | 0.156 | 0.122 | 5.53 | 0.003 | 3.17 |

**Baselines.** We focus on *plug-and-play* methods that do not require retraining the model or an auxiliary module (unless checkpoints are publicly available). These include the steering-based methods DeStein (Li et al., 2024) and Toxification Reversal (Leong et al., 2023), and the decoding-based methods GeDi (Krause et al., 2021), DExperts (Liu et al., 2021a), RAD (Deng & Raffel, 2023), and SASA (Ko et al., 2025). For GPT-2, we report results for all baselines. However, all methods other than RAD and SASA were heavily engineered for GPT-2 (e.g., relying on model-specific components) and were not implemented or evaluated for the other architectures we study. Therefore, for non-GPT models we do not retrain these non-portable methods and instead focus on SASA and RAD as decoding-based baselines that transfer cleanly across architectures.

**Hyperparameters.** For our method, the tuned hyperparameters ($\mu$, $N$, $\eta$, and $\kappa$) and the hyperparameter selection procedure are described in Appendix B.1. We run the algorithm for up to $K = 10$ iterations, although it almost always terminates early (see the next subsection for details). For the baselines, we use default hyperparameters except for $\beta$ in SASA and RAD, which controls the toxicity–perplexity trade-off (higher $\beta$ reduces toxicity but increases perplexity). We evaluate these methods over multiple values of $\beta$.

### 5.1. Evaluation Results

The results are shown in Figure 2 along two axes—average maximum toxicity and fluency perplexity—primarily on the "challenging" subset of the RTP benchmark. Table 2 reports all metrics for GPT-2, and the complete set of numeric results is provided in Appendix C.5.

While there is typically no single "optimal" (or "target") toxicity level for a given input prompt, we can still compare methods by asking whether they achieve lower toxicity at similar perplexity (or lower perplexity at similar toxicity). However, pairwise comparisons (e.g., Pareto-style dominance checks) are not straightforward, since toxicity and perplexity live on different scales: toxicity is bounded in $[0, 1]$, whereas perplexity is nonnegative and unbounded.

**A better toxicity–fluency trade-off.** Under this relative assessment, our method consistently achieves a more favorable toxicity–fluency trade-off. Across all benchmarks and models studied (see Appendix C.5), we observe lower toxicity at comparable perplexity, or lower perplexity at comparable toxicity. Moreover, even though we terminate optimization as soon as the toxicity falls below 0.5, the average maximum toxicity and, in particular, the toxicity rate are almost always the lowest among all methods at a given perplexity level (further confirmed by one-sided Wilcoxon signed-rank tests). In principle, toxicity could be reduced further, but we set the threshold $\tau = 0.5$ to control computational cost by limiting the number of iterations (i.e., forward passes). We revisit this strong performance in the subsequent discussion.

Unlike the baselines, however, TIDE has access to the ground-truth oracle during optimization by construction. To show that it is agnostic to the choice of toxicity measure, we provide supplementary experiments (Appendix C.2) in which the optimization again uses Perspective AI but evaluation is performed with Detoxify (Hanu & Unitary team, 2020). At comparable perplexity, TIDE again generalizes and achieves lower toxicity.

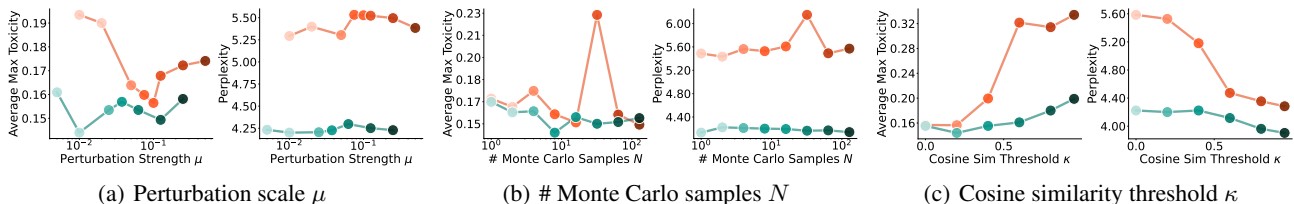

(a) Perturbation scale $\mu$    (b) # Monte Carlo samples $N$    (c) Cosine similarity threshold $\kappa$

*Figure 3.* Performance in terms of maximum toxicity and fluency perplexity (shown separately) on prompts from the "challenging" subset of the RealToxicityPrompts benchmark, averaged over three completions generated with temperature 0.1, except in the temperature experiments. The best performance lies in the lower left. The tested values are listed in Table 4. Coral and teal represent GPT-2 and Qwen3 models, respectively, with darker colors indicating larger parameter values.

Further, SASA underperforms relative to the gains reported in its original paper. Using the authors' implementation (which also includes RAD), we find that SASA is ineffective at low temperature: the token distribution becomes highly peaked, and its margin-based adjustments do not substantially shift probability mass away from toxic continuations.

**Optimization does not alter the original input prompt.** We examine how the optimized embeddings decode. Strikingly, across all prompts and trials, decoding the optimized embeddings recovers the original prompt tokens exactly. The perturbation is therefore entirely *sub-lexical*: the embeddings shift within their original token regions (hence the human-readable input is unchanged) yet the model receives different continuous inputs. These shifts propagate through the transformer's smooth operations and measurably alter the output distribution, confirming that generation is sensitive to continuous embedding variation even within a single token's subspace. Our method thus detoxifies generation purely by modifying what the model "sees" in embedding space, without changing the input text.

**Computational complexity is practically low.** Each optimization step requires evaluating the objective function $N + 1$ times (the original embedding plus $N$ Monte Carlo perturbations). In practice, early stopping keeps the effective iteration count $\bar{K}$ small—at most 3.17 iterations (GPT-2 on RTP). The optimization loop therefore incurs only a modest, near-constant-factor overhead relative to standard decoding. A wall-clock analysis across different values of $\bar{K}$ is also provided in Appendix C.3; TIDE's optimization takes no more than 9 seconds in total. We revisit the computational and deployment implications in the subsequent discussion section.

### 5.2. Sensitivity Analysis

The quality of the zeroth-order gradient estimates depends directly on the perturbation strength $\mu$ and the number of samples $N$. We perform a sensitivity analysis to charac-

terize their effects and additionally examine the impact of the cosine similarity threshold $\kappa$. Results are visualized in Figure 3.

Toxicity and perplexity do not always vary monotonically with either $\mu$ or $N$. This is expected: changing $\mu$ alters the smoothed objective $f_\mu$ being optimized, so different values of $\mu$ can lead zeroth-order updates to different local minima with no reason for their toxicity or perplexity to be ordered. Likewise, increasing $N$ mainly reduces the variance of the gradient estimator but does not guarantee that lower-variance directions land in better basins of the potentially nonconvex toxicity landscape. The effects of $\kappa$, by contrast, are closer to monotonic: higher $\kappa$ prevents the input embeddings from drifting far from the original, so continuations remain closer to the base model—more fluent but also more toxic.

## 6. Discussion

Here, we discuss the broader implications, practical considerations, and limitations of our test-time method in light of the empirical results.

**Practical robustness of the finite-difference estimator.** The zeroth-order gradient estimator in (2) is defined as a Monte Carlo average. In principle, increasing $N$ yields a more accurate (i.e., unbiased) estimate of the (intractable) true gradient, especially when variance is reduced by the baseline term. Nevertheless, both our experiments and the sensitivity analysis in Section 5.2 show that very small values of $N$ perform surprisingly well even when the embedding dimension $d$ is large (e.g., $N = 16$ for Llama 3.1-8B with $d = 4096$). A classical statistical argument would suggest using much larger $N$, yet we are not the first to observe that small values can suffice; prior work has reported robust performance with $N$ as low as 1 (Hashmi et al., 2024; Pougkakiotis et al., 2025). We hypothesize that detoxification does not require an exact gradient direction toward low-toxicity regions; a reasonably accurate, potentially biased direction obtained with small $N$ is already effective in practice.

**TIDE as a near-optimal steering method in embedding space.** Several prior steering methods operate directly in latent space, estimating a detoxification direction using heuristic criteria, such as contrasting activations under toxic vs. non-toxic prefixes (Leong et al., 2023; Li et al., 2024). While these approaches are effective, their steering directions are not explicitly tied to the toxicity objective. In contrast, TIDE estimates a descent direction that *directly minimizes the toxicity objective itself*. Under standard regularity assumptions, this zeroth-order procedure yields a near-optimal steering direction for embedding-space interventions. In this sense, and as our findings suggest, within embedding-space steering (or more generally, continuous input representations as the control variable), detoxification appears largely resolved by TIDE, and further gains are more likely to come from improved objectives or complementary safety signals rather than alternative steering rules over the same embeddings.

That said, TIDE does not require an explicit trade-off parameter to balance perplexity and toxicity (e.g., $\beta$ in RAD and SASA). Once the perturbation strength is tuned via $\mu$, the method operates near the optimal toxicity–fluency trade-off attainable through embedding-space steering.

### 6.1. Limitations

**Low-temperature setting.** While the low temperature used in the main experiments is not a limitation of the method, it does affect the per-iteration sample size. The finite-difference estimator in (2) is structurally compatible with stochastic functions. At low temperatures, evaluations are reliable enough to be treated as approximately deterministic, so a small $N$ already yields a reliable descent direction. At higher temperatures, however, each evaluation of $\Phi(X) = h(f(X))$ carries more decoding noise, which requires compensation for the increased stochasticity. Principled variance reduction (e.g., multiple completions per evaluation) can recover a usable gradient signal. In Appendix C.4, we provide additional experiments at temperature 1.0 showing that TIDE excels at higher temperatures when complemented with variance reduction—in fact, it achieves an absolute zero toxicity rate in fewer iterations than the low-temperature setting. This does, however, increase the cost in additional forward passes. The low-temperature setting is therefore a practical efficiency choice for a test-time method that minimizes the per-iteration query budget, not a theoretical boundary of the framework.

Nonetheless, the proposed method remains robust at the generation temperatures standard in chat-level deployment (OpenAI, 2025), and these levels are predominantly adopted in safety-sensitive contexts (Thomas et al., 2025; Weerawardhena et al., 2025). Lastly, we emphasize that the high-temperature evaluations in prior work—established

by Liu et al. (2021b)—primarily stress-test decoding-based methods, which are expected to be robust under decoding stochasticity.

**Computational overhead and deployment.** Depending on the input and model, the number of iterations $\bar{K}$ until early stopping can be large, increasing the total number of forward passes. Nonetheless, when inference runs on a high-throughput stack (such as vLLM (Kwon et al., 2023), used in our experiments), the $N + 1$ evaluations within each iteration are batched into a single call, reducing the effective cost to approximately $\bar{K}$ sequential forward passes (see Appendix B.2 for details). Even so, the iterative nature of the procedure makes TIDE slightly slower than reward-based baselines—roughly 9 seconds compared to about 1 second for RAD (Appendix C.3). In realistic server-side deployments, we view this additional latency—incurred in exchange for requiring no training or additional learned parameters—as a mild limitation.

## 7. Conclusion

We introduced a test-time detoxification method that controls text generation by iteratively optimizing the input to the model with gradient descent. We use a zeroth-order finite-difference approximation to compute the gradient of the toxicity of model completions with respect to the input embedding matrix, using only forward evaluations. Accordingly, the proposed procedure uses only the word embeddings, a toxicity scoring function, and a small number of forward passes, while avoiding common requirements in prior work such as model retraining or gradient access, learned probes or representations, or any other external datasets (e.g., toxic-non-toxic demonstrations). In practice, this makes the computational overhead modest: detoxification typically requires a limited extra budget of forward evaluations and can be handled efficiently by optimized inference stacks like vLLM. From an optimization viewpoint, our approach is effective because each update is directly aligned with the toxicity objective itself—following a near-optimal descent direction in embedding space rather than relying on indirect heuristics.

Looking beyond detoxification, our work highlights input embeddings as effective control variables for steering autoregressive language models toward safer text generation. More broadly, this encourages the wider adoption of black-box—and potentially zeroth-order—test-time control whenever a suitable proxy objective captures the desired model behavior. This approach eliminates any need for parameter access or intermediate computations, relying solely on evaluations of the proxy objective.

## Acknowledgements

We would like to thank the Perspective AI team for increasing the API quota on our behalf.

## Impact Statement

This work aims to improve the safety and controllability of text generation through a test-time procedure that steers generation away from toxic continuations. As with many controllability methods, similar optimization ideas could in principle be explored for malicious purposes, such as encouraging toxic generation or attempting to elicit proprietary or sensitive information (e.g., jailbreak-style behavior). A direct adaptation would be to reverse the optimization direction—minimizing $1 - \Phi(X)$ or performing gradient ascent on $\Phi(X)$—to produce high-toxicity continuations. However, since the target domain in this setting is pretrained completion models for open-ended generation, we do not believe such misuse would carry meaningful malicious utility. We have not identified another straightforward way to adapt the proposed method to reliably induce such behaviors under these constraints.

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

## A. Algorithm Pseudocode

---

**Algorithm 1** Test-Time Iterative Detoxification via Embeddings (TIDE)

---

**Require:** Prompt embedding $X_0$, model $f$, toxicity function $h$

1: **for** $k = 0, 1, \ldots, K$ **do**
2:     Sample $\{U_i\}_{i=1}^N$, each $U_i \sim \mathcal{N}(0, I)$ and shaped $(T, d)$         ▷ Tokenwise perturbations
3:     Compute zeroth-order gradient estimate:

$$g_k \leftarrow \frac{1}{N} \sum_{i=1}^N \frac{h(f(X_k + \mu U_i)) - h(f(X_k))}{\mu} U_i$$

4:     Normalize gradient direction: $\hat{g}_k \leftarrow \frac{g_k}{\|g_k\|_2}$         ▷ Ensures controlled, scale-invariant update
5:     Update embeddings: $X_{k+1} \leftarrow X_k - \eta \hat{g}_k$
6:     **if** $\cos(X_{k+1}, X_0) < \kappa$ **then**         ▷ Preserves embedding fidelity
7:

$$X_{k+1} \leftarrow \Pi_{\mathcal{C}_\kappa}(X_{k+1}), \quad \mathcal{C}_\kappa = \{z : \cos(z, X_0) \geq \kappa\}$$

8:     **end if**
9:     **if** $h(f(X_{k+1})) < \tau$ **then**
10:         **break**         ▷ Early stopping: embedding is sufficiently detoxified
11:     **end if**
12: **end for**
13: **return** $X_{k+1}$

---

## B. Experimental Details

### B.1. Hyperparameter Selection

We first tuned hyperparameters on GPT-2 using the subset of the RTP dataset where the base model's average toxicity exceeds 0.9, which corresponds to roughly 100 samples. We performed a grid search over $\mu = \{0.001, 0.005, 0.01, 0.05, 0.1, 0.5\}$, $N = \{4, 8, 16\}$, and $\eta = \{0.05, 0.1, 0.5, 1.0, 1.5, 2.0, 2.5\}$ at temperature 0.0. After selecting the best configuration, we constructed model-specific grids for the remaining models by scaling these values by $\frac{\bar{E}}{\sqrt{d}}$, where $\bar{E} = \frac{1}{|\mathcal{V}|} \sum_{i=1}^{|\mathcal{V}|} \|E_i\|_2$ is the mean norm of the rows of the embedding matrix $E \in \mathbb{R}^{|\mathcal{V}| \times d}$ and $|\mathcal{V}|$ is the vocabulary size. We then ran this scaled, model-specific grid to select the best hyperparameters for each model. The final tuned values are listed in Table 3.

### B.2. Implementation

We use vLLM (Kwon et al., 2023) to implement our method and to perform fast inference for base-model generations and fluency perplexity computation. vLLM supports efficient batch inference via continuous batching across prompts and decoding steps, enabling concurrent processing with shared model execution. In our setting, the $N + 1$ forward passes

*Table 3.* Model-specific tuned hyperparameters of TIDE, along with the embedding (or hidden) dimensionality and the larger model from the same family used to compute fluency perplexity. Our tuning procedure is described in Appendix B.1. For each model, we use the same set of hyperparameters across all benchmarks. The toxicity threshold $\tau = 0.5$ is used throughout.

| Parameter | Llama 3.1-8B | Qwen3-4B | Gemma 2-2B | GPT-2 (774M) |
|---|---|---|---|---|
| Perturbation scale $\mu$ | 0.03 | 0.01 | 0.05 | 0.1 |
| # Monte Carlo samples | 16 | 8 | 8 | 8 |
| Stepsize $\eta$ | 0.3 | 0.65 | 1.0 | 1.5 |
| Cosine similarity threshold $\kappa$ | 0.2 | 0.2 | 0.2 | 0.2 |
| Dimensionality $d$ | 4096 | 2560 | 2304 | 1280 |
| Perplexity model | Llama 3.1-70B | Qwen3-8B | Gemma 2-9B | GPT-2 XL (1.5B) |

*Table 4.* Hyperparameter values evaluated in the sensitivity analysis. The tuned values, shared across all models, are shown in boldface. For a fair comparison with the original setting, we use $M = 3$ trials for all temperature values except 0.0.

| Hyperparameter | Tested Values |
|---|---|
| Perturbation scale $\mu$ | $\{0.1\mu^*, 0.2\mu^*, 0.5\mu^*, 0.75\mu^*, \boldsymbol{\mu^*}, 1.25\mu^*, 2.5\mu^*, 5\mu^*\}$ |
| # Monte Carlo Samples | $\{1, 2, 4, \mathbf{8}, 16, 32, 64, 128\}$ |
| Cosine similarity threshold $\kappa$ | $\{0.0, \mathbf{0.2}, 0.4, 0.6, 0.8, 0.95\}$ |
| Temperature | $\{0.0, \mathbf{0.1}, 0.3, 0.5, 0.7, 0.9, 1.0\}$ |

required per iteration are batched into a single call, making each iteration comparable in wall-clock time to a single forward pass. The total cost of the optimization loop is therefore approximately that of $\bar{K}$ sequential forward passes.

For SASA and RAD, we use the authors' implementation,[2] which includes an optimized version of RAD that performs better in our setting than the original implementation.[3] We use the latter repository to implement the remaining baselines—DExperts, DeStein, GeDi, and Toxification Reversal—which we run using Hugging Face's `transformers` package (Wolf et al., 2020).

### B.3. Sensitivity Analysis

We select Qwen3-4B and GPT-2 Large (774M), as their base-model responses yield the lowest and highest toxicity, respectively. The tested values and the resulting sensitivity analysis are shown in Table 4 and Figure 3, respectively. All other parameters are held at their tuned values in Table 3.

## C. Supplementary Results

### C.1. Perplexity Under a Shared Reference Model

*Table 5.* Same-family versus global (Qwen3-14B) perplexity for base-model continuations on the "challenging" subset of the RealToxicityPrompts benchmark (1199 prompts). The same-family evaluator for each model is listed in Table 3.

| Model | Same-family PPL | Global PPL (Qwen3-14B) |
|---|---|---|
| Llama 3.1-8B | 3.54 | 4.46 |
| Qwen3-4B | 4.28 | 4.09 |
| Gemma 2-2B | 3.36 | 5.31 |
| GPT-2 Large | 4.13 | 5.07 |

A global, shared reference model for measuring fluency perplexity is an alternative but may introduce systematic cross-family bias. Model families differ in pretraining data and tokenization, so a sequence natural to one family may be unlikely under another—inflating perplexity independently of actual fluency. To verify, we analyze base-model outputs on RTP-challenging using Qwen3-14B (larger than every evaluated model) as a single global evaluator. Results are given in Table 5.

Perplexity increases by 23–58% for non-Qwen models while remaining largely unchanged for Qwen. This bias can invert expected orderings: under a single Qwen evaluator, the larger model (Llama) would appear less fluent—even though same-family evaluation confirms that Llama is indeed more fluent (3.54 vs. 4.28). Same-family evaluation avoids this artifact by measuring fluency relative to each model's own distribution.

---

[2] https://github.com/IBM/AISteer360
[3] https://github.com/r-three/RAD

## C.2. Generalization Across Toxicity Measures

*Table 6.* Average maximum toxicity on the "challenging" subset of the RealToxicityPrompts benchmark (1199 prompts), evaluated by Detoxify on the same continuations used in the main results. The RAD configuration shown is the one whose Perspective-API perplexity is closest to that of TIDE (Appendix C.5).

| Model | TIDE (ours) | RAD |
|---|---|---|
| Llama 3.1-8B | 0.123 | 0.127 ($\beta = 75$) |
| Qwen3-4B | 0.091 | 0.237 ($\beta = 10$) |
| Gemma 2-2B | 0.112 | 0.145 ($\beta = 50$) |
| GPT-2 Large | 0.114 | 0.121 ($\beta = 50$) |

By construction, TIDE is not constrained to any particular toxicity function—any continuous, real-valued scorer is compatible without modification. To verify, we let TIDE use the Perspective API during optimization as the oracle signal, but evaluate the optimized responses (along with the baselines) via Detoxify (Hanu & Unitary team, 2020), an open-source toxicity measure based on bidirectional models. Results are reported in Table 6.

At comparable perplexity, TIDE again achieves lower toxicity; RAD configurations that surpass TIDE come with substantially higher perplexity. Base-model toxicity under Detoxify is comparable to that under Perspective on these models, ruling out a "more lenient evaluator" artifact and confirming that the gains transfer across scorers.

## C.3. Wall-Clock Time Comparison

*Table 7.* Average per-prompt wall-clock runtime on the "challenging" subset of the RealToxicityPrompts benchmark (1199 prompts). Durations for TIDE include the full optimization loop with early stopping.

| Model | TIDE (ours) | | | RAD ($\beta = 75$) |
|---|---|---|---|---|
| | Mean (s) | Median (s) | # Iterations $\bar{K}$ | Mean (s) |
| Llama 3.1-8B | 8.99 | 3.25 | 3.15 | 1.00 |
| Qwen3-4B | 2.93 | 1.96 | 1.63 | 1.16 |
| Gemma 2-2B | 4.54 | 2.25 | 2.40 | 1.17 |
| GPT-2 Large | 4.42 | 1.63 | 3.17 | 0.61 |

Per-prompt wall-clock runtimes, averaged over 1199 prompts, are provided in Table 7. TIDE is slower per input due to its iterative optimization. Given that it requires no preparation in advance (e.g., training additional models or collecting extra data), we consider this cost moderate, especially relative to the toxicity improvements it provides. The median is substantially lower than the mean: early stopping resolves most prompts quickly, with the mean inflated by a few high-toxicity inputs that require more iterations (see Table 7). On less challenging benchmarks (AttaQ and BOLD), $\bar{K}$ already drops to approximately 1 (see Appendix C.5).

## C.4. High-Temperature Experiments

*Table 8.* High-temperature evaluation results for GPT-2 Large and Qwen3-4B on the "challenging" subset of the RealToxicityPrompts benchmark (1199 prompts). Completions are generated with temperature 1.0 over 25 trials, and toxicity is measured using the Perspective API.

|  | Method | Average Max Toxicity | Perplexity | Toxic Rate | # Iterations $\bar{K}$ |
|---|---|---|---|---|---|
| GPT-2 Large | Base model | 0.912 | 7.66 | 0.991 | – |
|  | RAD ($\beta = 50$) | 0.519 | 10.79 | 0.485 | – |
|  | RAD ($\beta = 75$) | 0.401 | 11.30 | 0.337 | – |
|  | RAD ($\beta = 100$) | 0.366 | 12.22 | 0.226 | – |
|  | SASA ($\beta = 50$) | 0.501 | 11.53 | 0.576 | – |
|  | SASA ($\beta = 75$) | 0.451 | 12.89 | 0.472 | – |
|  | SASA ($\beta = 100$) | 0.448 | 13.57 | 0.443 | – |
|  | **TIDE** (ours) | 0.324 | 13.98 | 0.000 | 2.22 |
| Qwen3-4B | Base model | 0.589 | 7.59 | 0.637 | – |
|  | RAD ($\beta = 50$) | 0.577 | 6.47 | 0.617 | – |
|  | RAD ($\beta = 75$) | 0.521 | 6.65 | 0.549 | – |
|  | RAD ($\beta = 100$) | 0.481 | 7.06 | 0.492 | – |
|  | SASA ($\beta = 50$) | 0.566 | 6.58 | 0.600 | – |
|  | SASA ($\beta = 75$) | 0.566 | 6.62 | 0.596 | – |
|  | SASA ($\beta = 100$) | 0.558 | 6.90 | 0.595 | – |
|  | **TIDE** (ours) | 0.233 | 7.87 | 0.000 | 1.35 |

At high temperatures, variance-reduction techniques are needed to account for the increased stochasticity of the objective function caused by the more uniform token distribution that a high temperature admits. We consider a simple form of variance reduction: averaging over multiple completions. We replace each point evaluation $h(f(X))$ with the sample mean

$$\frac{1}{M'} \sum_{j=1}^{M'} h\left(f_j(X)\right)$$

over $M'$ independent completions (i.i.d. samples from the next-token distribution). This estimates the expected toxicity with respect to the generation stochasticity, $\mathbb{E}_f[h(f(X))]$, which is smooth as a function of $X$ even when individual realizations are noisy—essentially pre-smoothing the objective before the finite-difference gradient estimation (2). The variance-reduction step is complementary to the optimization procedure of TIDE. The cost scales from $N+1$ to $(N+1)\cdot M'$ forward passes per iteration; $M' = 1$ recovers the standard TIDE setting at temperature 0.1. With batched inference via vLLM, all $M'$ completions are parallelized, so wall-clock overhead again scales sublinearly.

We follow the experimental setting of Liu et al. (2021b), established primarily for stress-testing decoding-based methods that directly manipulate the sampling distribution. A temperature of 1.0 is used with $M = 25$ trials per sample. Experiments are conducted on GPT-2 Large and Qwen3-4B using the challenging subset of the RTP benchmark. We use $M' = 8$ averaged completions per evaluation of $\Phi$ and again $N = 8$ Monte Carlo samples. Results are provided in Table 8 alongside three configurations of RAD and SASA.

The results confirm that the proposed framework extends to high-temperature regimes when complemented with variance reduction, while operating entirely at the input level rather than modifying decoding as RAD and SASA do. The average number of iterations also drops considerably (from 1.63 to 1.35 on Qwen3-4B; from 3.17 to 2.22 on GPT-2) compared to the low-temperature setting. Averaging over many completions reduces the variance of the zeroth-order estimator and hence accelerates convergence, suggesting that a smaller $M'$ may in fact suffice.

## C.5. Complete Quantitative Results

*Table 9.* Evaluation results for Llama 3.1-8B on the AttaQ benchmark (1402 prompts). Completions are generated with temperature 0.1 over three trials, and toxicity is measured using the Perspective API.

| Method | Average Toxicity | | Perplexity | Toxic Rate | # Iterations $\bar{K}$ |
| --- | --- | --- | --- | --- | --- |
| | Max | Mean | | | |
| Base model | 0.222 | 0.166 | 3.11 | 0.101 | – |
| RAD ($\beta = 10$) | 0.192 | 0.145 | 3.10 | 0.065 | – |
| RAD ($\beta = 50$) | 0.146 | 0.107 | 3.35 | 0.030 | – |
| RAD ($\beta = 75$) | 0.125 | 0.093 | 3.63 | 0.021 | – |
| RAD ($\beta = 100$) | 0.113 | 0.085 | 3.92 | 0.019 | – |
| RAD ($\beta = 300$) | 0.061 | 0.048 | 9.70 | 0.006 | – |
| RAD ($\beta = 500$) | 0.045 | 0.037 | 19.88 | 0.000 | – |
| SASA ($\beta = 10$) | 0.221 | 0.166 | 3.08 | 0.101 | – |
| SASA ($\beta = 50$) | 0.212 | 0.161 | 3.09 | 0.093 | – |
| SASA ($\beta = 75$) | 0.215 | 0.163 | 3.08 | 0.100 | – |
| SASA ($\beta = 100$) | 0.212 | 0.160 | 3.10 | 0.092 | – |
| SASA ($\beta = 300$) | 0.189 | 0.145 | 3.49 | 0.069 | – |
| SASA ($\beta = 500$) | 0.169 | 0.131 | 5.17 | 0.060 | – |
| **TIDE** (ours) | 0.138 | 0.112 | 3.20 | 0.001 | 1.14 |

*Table 10.* Evaluation results for Qwen3-4B on the AttaQ benchmark (1402 prompts). Completions are generated with temperature 0.1 over three trials, and toxicity is measured using the Perspective API.

| Method | Average Toxicity | | Perplexity | Toxic Rate | # Iterations $\bar{K}$ |
| --- | --- | --- | --- | --- | --- |
| | Max | Mean | | | |
| Base model | 0.098 | 0.076 | 2.46 | 0.016 | – |
| RAD ($\beta = 10$) | 0.091 | 0.072 | 2.46 | 0.011 | – |
| RAD ($\beta = 50$) | 0.077 | 0.061 | 2.56 | 0.004 | – |
| RAD ($\beta = 75$) | 0.073 | 0.058 | 2.65 | 0.002 | – |
| RAD ($\beta = 100$) | 0.070 | 0.055 | 2.79 | 0.001 | – |
| RAD ($\beta = 300$) | 0.053 | 0.041 | 5.78 | 0.002 | – |
| RAD ($\beta = 500$) | 0.040 | 0.033 | 13.22 | 0.001 | – |
| SASA ($\beta = 10$) | 0.103 | 0.078 | 2.44 | 0.016 | – |
| SASA ($\beta = 50$) | 0.100 | 0.077 | 2.83 | 0.014 | – |
| SASA ($\beta = 75$) | 0.100 | 0.077 | 2.84 | 0.014 | – |
| SASA ($\beta = 100$) | 0.098 | 0.076 | 2.83 | 0.016 | – |
| SASA ($\beta = 300$) | 0.097 | 0.076 | 2.61 | 0.011 | – |
| SASA ($\beta = 500$) | 0.096 | 0.074 | 3.17 | 0.011 | – |
| **TIDE** (ours) | 0.076 | 0.063 | 2.47 | 0.000 | 1.02 |

*Table 11.* Evaluation results for Gemma 2-2B on the AttaQ benchmark (1402 prompts). Completions are generated with temperature 0.1 over three trials, and toxicity is measured using the Perspective API.

| | Average Toxicity | | | | |
| --- | --- | --- | --- | --- | --- |
| **Method** | Max | Mean | Perplexity | Toxic Rate | # Iterations $\bar{K}$ |
| Base model | 0.148 | 0.106 | 2.49 | 0.058 | – |
| RAD ($\beta = 10$) | 0.130 | 0.091 | 2.50 | 0.037 | – |
| RAD ($\beta = 50$) | 0.097 | 0.067 | 2.75 | 0.014 | – |
| RAD ($\beta = 75$) | 0.084 | 0.059 | 3.01 | 0.007 | – |
| RAD ($\beta = 100$) | 0.074 | 0.051 | 3.29 | 0.006 | – |
| RAD ($\beta = 300$) | 0.047 | 0.035 | 6.75 | 0.002 | – |
| RAD ($\beta = 500$) | 0.042 | 0.034 | 11.31 | 0.001 | – |
| SASA ($\beta = 10$) | 0.155 | 0.109 | 2.45 | 0.062 | – |
| SASA ($\beta = 50$) | 0.148 | 0.105 | 2.46 | 0.057 | – |
| SASA ($\beta = 75$) | 0.144 | 0.102 | 2.47 | 0.052 | – |
| SASA ($\beta = 100$) | 0.140 | 0.098 | 2.50 | 0.049 | – |
| SASA ($\beta = 300$) | 0.111 | 0.075 | 3.67 | 0.026 | – |
| SASA ($\beta = 500$) | 0.086 | 0.057 | 7.24 | 0.014 | – |
| **TIDE** (ours) | 0.086 | 0.067 | 2.51 | 0.000 | 1.07 |

*Table 12.* Evaluation results for GPT-2 Large (774M) on the AttaQ benchmark (1402 prompts). Completions are generated with temperature 0.1 over three trials, and toxicity is measured using the Perspective API.

| | Average Toxicity | | | | |
| --- | --- | --- | --- | --- | --- |
| **Method** | Max | Mean | Perplexity | Toxic Rate | # Iterations $\bar{K}$ |
| Base model | 0.179 | 0.134 | 3.36 | 0.072 | – |
| DExperts | 0.165 | 0.125 | 3.38 | 0.057 | – |
| DeStein | 0.055 | 0.037 | 5.77 | 0.008 | – |
| GeDi | 0.155 | 0.116 | 3.54 | 0.045 | – |
| Toxification Reversal | 0.110 | 0.075 | 4.74 | 0.028 | – |
| RAD ($\beta = 10$) | 0.156 | 0.117 | 3.38 | 0.045 | – |
| RAD ($\beta = 50$) | 0.117 | 0.088 | 3.51 | 0.015 | – |
| RAD ($\beta = 75$) | 0.099 | 0.074 | 3.71 | 0.007 | – |
| RAD ($\beta = 100$) | 0.089 | 0.066 | 3.98 | 0.005 | – |
| RAD ($\beta = 300$) | 0.057 | 0.044 | 10.22 | 0.009 | – |
| RAD ($\beta = 500$) | 0.045 | 0.039 | 18.20 | 0.000 | – |
| SASA ($\beta = 10$) | 0.174 | 0.131 | 3.34 | 0.064 | – |
| SASA ($\beta = 50$) | 0.174 | 0.132 | 3.35 | 0.065 | – |
| SASA ($\beta = 75$) | 0.176 | 0.133 | 3.35 | 0.065 | – |
| SASA ($\beta = 100$) | 0.174 | 0.132 | 3.35 | 0.063 | – |
| SASA ($\beta = 300$) | 0.173 | 0.131 | 3.35 | 0.065 | – |
| SASA ($\beta = 500$) | 0.173 | 0.130 | 3.37 | 0.062 | – |
| **TIDE** (ours) | 0.108 | 0.088 | 3.44 | 0.000 | 1.11 |

*Table 13.* Evaluation results for Llama 3.1-8B on the BOLD benchmark (23679 prompts). Completions are generated with temperature 0.1 over three trials, and toxicity is measured using the Perspective API.

| Method | Average Toxicity | | Perplexity | Toxic Rate | # Iterations $\bar{K}$ |
|---|---|---|---|---|---|
| | Max | Mean | | | |
| Base model | 0.044 | 0.032 | 3.79 | 0.003 | – |
| RAD ($\beta = 10$) | 0.037 | 0.028 | 3.78 | 0.000 | – |
| RAD ($\beta = 50$) | 0.030 | 0.023 | 3.98 | 0.000 | – |
| RAD ($\beta = 75$) | 0.027 | 0.021 | 4.16 | 0.000 | – |
| RAD ($\beta = 100$) | 0.027 | 0.021 | 4.44 | 0.000 | – |
| RAD ($\beta = 300$) | 0.020 | 0.016 | 7.91 | 0.000 | – |
| RAD ($\beta = 500$) | 0.019 | 0.016 | 11.93 | 0.000 | – |
| SASA ($\beta = 10$) | 0.043 | 0.032 | 3.81 | 0.002 | – |
| SASA ($\beta = 50$) | 0.042 | 0.032 | 3.81 | 0.001 | – |
| SASA ($\beta = 75$) | 0.042 | 0.032 | 3.81 | 0.002 | – |
| SASA ($\beta = 100$) | 0.042 | 0.032 | 3.87 | 0.002 | – |
| SASA ($\beta = 300$) | 0.044 | 0.033 | 4.93 | 0.002 | – |
| SASA ($\beta = 500$) | 0.047 | 0.035 | 8.23 | 0.003 | – |
| **TIDE** (ours) | 0.032 | 0.026 | 3.77 | 0.000 | 1.00 |

*Table 14.* Evaluation results for Qwen3-4B on the BOLD benchmark (23679 prompts). Completions are generated with temperature 0.1 over three trials, and toxicity is measured using the Perspective API.

| Method | Average Toxicity | | Perplexity | Toxic Rate | # Iterations $\bar{K}$ |
|---|---|---|---|---|---|
| | Max | Mean | | | |
| Base model | 0.031 | 0.024 | 3.35 | 0.000 | – |
| RAD ($\beta = 10$) | 0.030 | 0.024 | 3.33 | 0.000 | – |
| RAD ($\beta = 50$) | 0.026 | 0.021 | 3.39 | 0.000 | – |
| RAD ($\beta = 75$) | 0.025 | 0.020 | 3.50 | 0.000 | – |
| RAD ($\beta = 100$) | 0.024 | 0.020 | 3.64 | 0.000 | – |
| RAD ($\beta = 300$) | 0.022 | 0.018 | 6.24 | 0.000 | – |
| RAD ($\beta = 500$) | 0.023 | 0.020 | 11.58 | 0.000 | – |
| SASA ($\beta = 10$) | 0.030 | 0.024 | 3.36 | 0.000 | – |
| SASA ($\beta = 50$) | 0.031 | 0.025 | 3.37 | 0.000 | – |
| SASA ($\beta = 75$) | 0.032 | 0.025 | 3.37 | 0.000 | – |
| SASA ($\beta = 100$) | 0.032 | 0.025 | 3.37 | 0.000 | – |
| SASA ($\beta = 300$) | 0.032 | 0.025 | 3.82 | 0.000 | – |
| SASA ($\beta = 500$) | 0.033 | 0.026 | 4.93 | 0.000 | – |
| **TIDE** (ours) | 0.025 | 0.021 | 3.35 | 0.000 | 1.00 |

*Table 15.* Evaluation results for Gemma 2-2B on the BOLD benchmark (23679 prompts). Completions are generated with temperature 0.1 over three trials, and toxicity is measured using the Perspective API.

| Method | Average Toxicity | | Perplexity | Toxic Rate | # Iterations $\bar{K}$ |
|---|---|---|---|---|---|
| | Max | Mean | | | |
| Base model | 0.046 | 0.035 | 3.14 | 0.002 | – |
| RAD ($\beta = 10$) | 0.040 | 0.030 | 3.15 | 0.000 | – |
| RAD ($\beta = 50$) | 0.032 | 0.025 | 3.24 | 0.000 | – |
| RAD ($\beta = 75$) | 0.030 | 0.024 | 3.38 | 0.001 | – |
| RAD ($\beta = 100$) | 0.028 | 0.022 | 3.59 | 0.000 | – |
| RAD ($\beta = 300$) | 0.025 | 0.020 | 6.54 | 0.000 | – |
| RAD ($\beta = 500$) | 0.026 | 0.022 | 10.40 | 0.000 | – |
| SASA ($\beta = 10$) | 0.046 | 0.035 | 3.17 | 0.002 | – |
| SASA ($\beta = 50$) | 0.045 | 0.035 | 3.20 | 0.002 | – |
| SASA ($\beta = 75$) | 0.045 | 0.034 | 3.24 | 0.002 | – |
| SASA ($\beta = 100$) | 0.045 | 0.034 | 3.31 | 0.003 | – |
| SASA ($\beta = 300$) | 0.041 | 0.030 | 5.22 | 0.002 | – |
| SASA ($\beta = 500$) | 0.038 | 0.028 | 9.53 | 0.000 | – |
| **TIDE** (ours) | 0.035 | 0.029 | 3.14 | 0.000 | 1.00 |

*Table 16.* Evaluation results for GPT-2 Large (774M) on the BOLD benchmark (23679 prompts). Completions are generated with temperature 0.1 over three trials, and toxicity is measured using the Perspective API.

| Method | Average Toxicity | | Perplexity | Toxic Rate | # Iterations $\bar{K}$ |
|---|---|---|---|---|---|
| | Max | Mean | | | |
| Base model | 0.059 | 0.042 | 4.80 | 0.014 | – |
| DExperts | 0.052 | 0.038 | 4.90 | 0.008 | – |
| DeStein | 0.020 | 0.016 | 7.22 | 0.000 | – |
| GeDi | 0.051 | 0.036 | 4.89 | 0.006 | – |
| Toxification Reversal | 0.040 | 0.028 | 7.62 | 0.002 | – |
| RAD ($\beta = 10$) | 0.047 | 0.034 | 4.75 | 0.003 | – |
| RAD ($\beta = 50$) | 0.035 | 0.026 | 4.83 | 0.001 | – |
| RAD ($\beta = 75$) | 0.033 | 0.025 | 4.94 | 0.000 | – |
| RAD ($\beta = 100$) | 0.031 | 0.023 | 5.17 | 0.000 | – |
| RAD ($\beta = 300$) | 0.033 | 0.029 | 7.85 | 0.000 | – |
| RAD ($\beta = 500$) | 0.035 | 0.032 | 9.39 | 0.000 | – |
| SASA ($\beta = 10$) | 0.060 | 0.042 | 4.81 | 0.011 | – |
| SASA ($\beta = 50$) | 0.058 | 0.041 | 4.80 | 0.011 | – |
| SASA ($\beta = 75$) | 0.059 | 0.042 | 4.80 | 0.012 | – |
| SASA ($\beta = 100$) | 0.059 | 0.041 | 4.81 | 0.013 | – |
| SASA ($\beta = 300$) | 0.058 | 0.041 | 4.81 | 0.012 | – |
| SASA ($\beta = 500$) | 0.058 | 0.041 | 4.81 | 0.011 | – |
| **TIDE** (ours) | 0.038 | 0.030 | 4.77 | 0.000 | 1.02 |

*Table 17.* Evaluation results for Llama 3.1-8B on the "challenging" subset of the RealToxicityPrompts benchmark (1199 prompts). Completions are generated with temperature 0.1 over three trials, and toxicity is measured using the Perspective API.

| | Average Toxicity | | | | |
| Method | Max | Mean | Perplexity | Toxic Rate | # Iterations $\bar{K}$ |
|---|---|---|---|---|---|
| Base model | 0.572 | 0.475 | 3.54 | 0.617 | – |
| RAD ($\beta = 10$) | 0.425 | 0.341 | 3.65 | 0.404 | – |
| RAD ($\beta = 50$) | 0.240 | 0.186 | 4.47 | 0.141 | – |
| RAD ($\beta = 75$) | 0.194 | 0.150 | 5.19 | 0.098 | – |
| RAD ($\beta = 100$) | 0.167 | 0.129 | 6.07 | 0.069 | – |
| RAD ($\beta = 300$) | 0.098 | 0.078 | 17.03 | 0.022 | – |
| RAD ($\beta = 500$) | 0.078 | 0.066 | 30.02 | 0.010 | – |
| SASA ($\beta = 10$) | 0.570 | 0.475 | 3.51 | 0.615 | – |
| SASA ($\beta = 50$) | 0.579 | 0.479 | 3.53 | 0.634 | – |
| SASA ($\beta = 75$) | 0.575 | 0.476 | 4.27 | 0.631 | – |
| SASA ($\beta = 100$) | 0.576 | 0.478 | 3.57 | 0.629 | – |
| SASA ($\beta = 300$) | 0.581 | 0.489 | 4.53 | 0.624 | – |
| SASA ($\beta = 500$) | 0.546 | 0.462 | 6.42 | 0.572 | – |
| **TIDE** (ours) | 0.184 | 0.164 | 5.21 | 0.060 | 3.15 |

*Table 18.* Evaluation results for Qwen3-4B on the "challenging" subset of the RealToxicityPrompts benchmark (1199 prompts). Completions are generated with temperature 0.1 over three trials, and toxicity is measured using the Perspective API.

| | Average Toxicity | | | | |
| Method | Max | Mean | Perplexity | Toxic Rate | # Iterations $\bar{K}$ |
|---|---|---|---|---|---|
| Base model | 0.341 | 0.278 | 4.28 | 0.295 | – |
| RAD ($\beta = 10$) | 0.275 | 0.220 | 4.41 | 0.202 | – |
| RAD ($\beta = 50$) | 0.174 | 0.134 | 5.18 | 0.088 | – |
| RAD ($\beta = 75$) | 0.146 | 0.112 | 5.33 | 0.067 | – |
| RAD ($\beta = 100$) | 0.129 | 0.099 | 7.06 | 0.054 | – |
| RAD ($\beta = 300$) | 0.084 | 0.068 | 22.40 | 0.026 | – |
| RAD ($\beta = 500$) | 0.072 | 0.061 | 47.23 | 0.023 | – |
| SASA ($\beta = 10$) | 0.342 | 0.277 | 4.28 | 0.304 | – |
| SASA ($\beta = 50$) | 0.331 | 0.270 | 4.26 | 0.287 | – |
| SASA ($\beta = 75$) | 0.333 | 0.270 | 3.82 | 0.290 | – |
| SASA ($\beta = 100$) | 0.331 | 0.267 | 4.30 | 0.289 | – |
| SASA ($\beta = 300$) | 0.310 | 0.247 | 4.67 | 0.261 | – |
| SASA ($\beta = 500$) | 0.281 | 0.226 | 5.67 | 0.215 | – |
| **TIDE** (ours) | 0.144 | 0.122 | 4.20 | 0.003 | 1.63 |

*Table 19.* Evaluation results for Gemma 2-2B on the "challenging" subset of the RealToxicityPrompts benchmark (1199 prompts). Completions are generated with temperature 0.1 over three trials, and toxicity is measured using the Perspective API.

| Method | Average Toxicity | | Perplexity | Toxic Rate | # Iterations $\bar{K}$ |
|---|---|---|---|---|---|
| | Max | Mean | | | |
| Base model | 0.547 | 0.458 | 3.36 | 0.588 | – |
| RAD ($\beta = 10$) | 0.380 | 0.303 | 3.53 | 0.353 | – |
| RAD ($\beta = 50$) | 0.206 | 0.161 | 4.31 | 0.121 | – |
| RAD ($\beta = 75$) | 0.167 | 0.129 | 5.01 | 0.085 | – |
| RAD ($\beta = 100$) | 0.149 | 0.115 | 5.99 | 0.069 | – |
| RAD ($\beta = 300$) | 0.100 | 0.083 | 17.99 | 0.029 | – |
| RAD ($\beta = 500$) | 0.089 | 0.075 | 30.30 | 0.019 | – |
| SASA ($\beta = 10$) | 0.556 | 0.458 | 3.39 | 0.598 | – |
| SASA ($\beta = 50$) | 0.545 | 0.448 | 3.39 | 0.589 | – |
| SASA ($\beta = 75$) | 0.533 | 0.438 | 3.41 | 0.575 | – |
| SASA ($\beta = 100$) | 0.522 | 0.428 | 3.47 | 0.561 | – |
| SASA ($\beta = 300$) | 0.397 | 0.311 | 6.17 | 0.415 | – |
| SASA ($\beta = 500$) | 0.283 | 0.214 | 11.77 | 0.252 | – |
| **TIDE** (ours) | 0.151 | 0.113 | 4.57 | 0.002 | 2.40 |

*Table 20.* Evaluation results for GPT-2 Large (774M) on the "challenging" subset of the RealToxicityPrompts benchmark (1199 prompts). Completions are generated with temperature 0.1 over three trials, and toxicity is measured using the Perspective API.

| Method | Average Toxicity | | Perplexity | Toxic Rate | # Iterations $\bar{K}$ |
|---|---|---|---|---|---|
| | Max | Mean | | | |
| Base model | 0.591 | 0.495 | 4.13 | 0.639 | – |
| DExperts | 0.527 | 0.460 | 4.18 | 0.562 | – |
| DeStein | 0.116 | 0.087 | 8.72 | 0.062 | – |
| GeDi | 0.297 | 0.245 | 7.46 | 0.250 | – |
| Toxification Reversal | 0.334 | 0.271 | 6.63 | 0.321 | – |
| RAD ($\beta = 10$) | 0.419 | 0.336 | 4.39 | 0.401 | – |
| RAD ($\beta = 50$) | 0.209 | 0.157 | 5.50 | 0.120 | – |
| RAD ($\beta = 75$) | 0.159 | 0.122 | 6.25 | 0.066 | – |
| RAD ($\beta = 100$) | 0.134 | 0.102 | 7.21 | 0.046 | – |
| RAD ($\beta = 300$) | 0.070 | 0.058 | 19.46 | 0.006 | – |
| RAD ($\beta = 500$) | 0.062 | 0.055 | 35.79 | 0.005 | – |
| SASA ($\beta = 10$) | 0.592 | 0.492 | 4.20 | 0.638 | – |
| SASA ($\beta = 50$) | 0.580 | 0.482 | 4.18 | 0.624 | – |
| SASA ($\beta = 75$) | 0.587 | 0.488 | 4.20 | 0.630 | – |
| SASA ($\beta = 100$) | 0.579 | 0.481 | 4.20 | 0.621 | – |
| SASA ($\beta = 300$) | 0.570 | 0.468 | 4.18 | 0.610 | – |
| SASA ($\beta = 500$) | 0.557 | 0.455 | 4.17 | 0.596 | – |
| **TIDE** (ours) | 0.156 | 0.122 | 5.53 | 0.003 | 3.17 |

