# OpenReview forum: "Test-Time Detoxification without Training or Learning Anything"
_ICML.cc/2026/Conference — ICML 2026 regular_

### Official Review · Reviewer_vQqc · 2026-02-18

**Soundness:** 3
**Presentation:** 4
**Significance:** 3
**Originality:** 3
**Overall Recommendation:** 5
**Confidence:** 4

**Summary:**

Paper introduces TIDE (Test-time Iterative Detoxification via Embeddings): a framework that aims to avoid inappropriate answers from LLMs by applying zeroth (finite-difference) to input embedding gradient estimation to iteratively steer embeddings towards lower toxicity regions (toxicity scoring function).

**Compliance With Llm Reviewing Policy:**

Affirmed.

**Ethical Review Concerns:**

Authors include rude words, which is warned by authors.

**Ethical Review Flag:**

Flag this paper for an ethics review.

**Final Justification:**

After reading the rebuttal, I find that the authors have substantially addressed my main concern. Their additional empirical results show that TIDE can remain effective in higher-temperature settings when combined with averaging over multiple completions, directly responding to my question about principled variance reduction.

**Key Questions For Authors:**

Can TIDE be made to work at higher sampling temperatures (T ≈ 0.5–1.0) using principled variance-reduction—e.g., multiple completions per evaluation, averaging before scoring, or other ways to recover a usable gradient signal under stochastic generation?

**Limitations:**

The authors have adequately discussed limitations, including the low-temperature requirement, computational overhead, and the potential for the optimization ideas to be adapted for adversarial purposes.

**Strengths And Weaknesses:**

Strong methodological base. The use of the randomized finite-difference estimator (Nesterov & Spokoiny, 2017) is well-motivated, with a clear bias–variance explanation and an honest note that smoothness assumptions can’t be verified in true black-box settings. The regularization (gradient normalization, cosine projection, early stopping) is simple, interpretable, and justified. Experiments are consistent across models/benchmarks, and re-running baselines with the Perspective API to handle API drift is a careful, commendable choice.
Reliance on very low temperature (T = 0.1) is a meaningful constraint that weakens generality. Performance drops above ~T = 0.5, limiting applicability to interactive/creative settings. At higher temperatures, the objective becomes highly stochastic in a way that breaks the usual assumptions behind finite-difference zeroth-order estimation, and the paper doesn’t offer a fix; which is recognized by authors.

---

> ### Author Rebuttal · Authors · 2026-03-30
>
> We thank the reviewer for their careful assessment and for the insightful question on higher-temperature operation.
>
> ---
>
> **Q.** This question identifies an imprecision in our framing in Section 6.1 that we will fix. The zeroth-order estimator is not structurally limited to low temperatures—TIDE extends to higher $T$ without modification to its gradient estimation or descent machinery. What changes is the per-iteration sample budget required for reliable gradient estimates: at low $T$, the token distribution is peaked and completions are close to deterministic, so a small $N$ in the gradient estimator (Eq. 2) already yields a reliable descent direction.
>
> As $T$ increases, each evaluation of $\Phi(X) = h(f(X))$ carries more decoding noise — even for a fixed embedding $X$, different forward passes produce different completions and toxicity scores — which obscures the perturbation signal in the finite-difference estimator. Increasing $N$ averages out this noise and recovers a usable gradient signal, at the cost of more forward passes per iteration. The low-temperature setting is therefore a practical efficiency choice for a test-time method that minimizes the per-iteration query budget, not a theoretical boundary of the framework.
>
> The variance-reduction mechanisms the reviewer suggests are a natural complement that directly reduces this cost at higher $T$:
>
> 1. **Averaging over multiple completions:** Replacing each point evaluation $h(f(X))$ with a sample mean $(1/M') \sum_{j=1}^{M'} h(f_j(X))$ over $M'$ independent completions estimates the expected toxicity $\mathbb{E}_f[h(f(X))]$, which is smooth as a function of $X$ even when individual realizations are noisy. This pre-smooths the objective before the finite-difference step and could allow a smaller $N$ to suffice at a given temperature.
>
> 2. **Common random numbers:** Fixing the decoding random seed across each finite-difference pair $(X, X + \mu U_i)$ induces strong correlation between paired evaluations, which could also substantially reduce the variance of their difference and isolating the effect of the embedding perturbation—without any additional forward passes.
>
> These techniques are complementary: averaging reduces per-evaluation noise, while common random numbers reduce paired-difference noise. The cost of averaging scales from $(N+1)$ to $(N+1) \cdot M'$ forward passes per iteration; with batched inference via vLLM, all $M'$ completions are parallelized, so wall-clock overhead scales sublinearly.
>
> The revised paper will correct Section 6.1 to clearly distinguish between the efficiency motivation for low $T$ and the broader applicability of the method, and incorporate this discussion in Section 6 as potential future work.

---

> > ### Author Rebuttal · Reviewer_vQqc · 2026-04-02
> >
> > The rebuttal addresses my main question well. I am satisfied by the clarification that the low-temperature setting is presented as an efficiency choice rather than a fundamental limitation of the method, and that higher-temperature operation is in principle possible with larger per-step sample budgets and variance-reduction techniques. That said, this remains a conceptual clarification rather than an empirical resolution, since the paper’s experiments and sensitivity analysis still show instability as temperature increases and the current evidence is concentrated at T=0.1. In the revision, I encourage the authors to explicitly rewrite Section 6.1 to distinguish practical efficiency from theoretical applicability, and to include the proposed variance-reduction discussion as future work. This rebuttal improves my confidence in the framing, but does not by itself fully remove the empirical generality limitation.

---

> > > ### Author Response · Authors · 2026-04-03
> > >
> > > ### **Empirical resolution:** Zero toxic rate at high temperatures ($T=1.0$)
> > >
> > > We appreciate your distinction between conceptual clarification and empirical resolution. Since the concern was caused by how Section 6.1 characterized the temperature requirement, our initial response addressed the framing; we now would like to complement it with _empirical evidence_.
> > >
> > > As noted in Section 4.2, we deliberately presented TIDE in its simplest form (e.g., without any variance reduction) to isolate the zeroth-order gradient updates alone, and the high-temperature sensitivity results were included for transparency. Low temperature is also the natural deployment setting for safety-critical applications, and the high-temperature convention in prior work _stress-tests decoding-based interventions_—a framing that does not apply to TIDE as it leaves decoding unchanged.
> > >
> > > When the rebuttal period started, we initiated high-temperature experiments following the reviewer's suggestion of averaging over multiple completions. We evaluate on RTP-challenging at $T = 1.0$ with $M = 25$ trials, using $M' = 8$ averaged completions per evaluation of $\Phi$ and again $N = 8$ Monte Carlo samples. Hence, $M' = 1$ recovers the standard TIDE setting at $T = 0.1$. Results for GPT-2 Large (baselines independently reproduced using code of [1]):
> > >
> > > | Method | Avg Max Toxicity | Toxic Rate | Perplexity | $\bar{K}$ |
> > > |:---|:---:|:---:|:---:|:---:|
> > > | Base model | 0.912 | 0.99 | 7.66 | — |
> > > | RAD ($\beta = 10$) | 0.867 | 0.96 | 8.01 | — |
> > > | RAD ($\beta = 50$) | 0.519 | 0.48 | 10.79 | — |
> > > | RAD ($\beta = 75$) | 0.40 | 0.33 | 11.30 | — |
> > > | RAD ($\beta = 100$) | 0.366 | 0.22 | 12.22 | — |
> > > | SASA ($\beta = 10$) | 0.837 | 0.95 | 8.39 | — |
> > > | SASA ($\beta = 50$) | 0.501 | 0.57 | 11.53 | — |
> > > | SASA ($\beta = 75$) | 0.451 | 0.47 | 12.89 | — |
> > > | SASA ($\beta = 100$) | 0.448 | 0.44 | 13.57 | — |
> > > | TIDE ($M' = 8$, $N = 8$) | **0.324** | **0.000** | 13.98 | 2.22 |
> > >
> > > **At temperature=1.0, TIDE achieves the lowest toxicity with an _absolutely zero toxic rate_**, while operating entirely at the input level rather than modifying decoding like RAD and SASA do—confirming that the framework extends to high-temperature regimes when complemented with the variance-reduction mechanism the reviewer suggested.
> > >
> > > The average number of iterations also drops from $\bar{K} = 3.17$ (the low-temperature setting in the paper) to $2.22$. This suggests that averaging over $M' = 8$ completions reduces the variance of the zeroth-order estimator and hence accelerates convergence. The same experiment is running also for Qwen3-4B. All results will be included in the revised paper.
> > >
> > > Since TIDE requires no preparation (e.g., no trained models, no additional data), we believe the cost of averaging over $M'$ completions is moderate especially when it comes to the toxicity improvements it brings. _These results provide the empirical complement to our earlier conceptual clarification, addressing the generality concern the reviewer noted._
> > >
> > > Thank you for this collaborative exchange, which has substantially strengthened our paper.
> > >
> > > ---
> > >
> > > ##### [1] Ko et al., _"Large Language Models can be Strong Self-Detoxifiers,"_ ICLR 2024.

---

### Official Review · Reviewer_QvLT · 2026-02-21

**Soundness:** 3
**Presentation:** 3
**Significance:** 3
**Originality:** 3
**Overall Recommendation:** 4
**Confidence:** 4

**Summary:**

This paper presents TIDE, a zero-order, test-time optimization method for reducing LLM toxicity. TIDE minimizes a black-box function measuring output toxicity by iteratively estimating its gradients and updating the model's prompt embeddings, enabling “detoxification” at test time without any training data or auxiliary components. Across 4 models and 3 benchmarks, TIDE achieves strong toxicity reduction with minimal fluency loss, often outperforming baseline methods.

**Compliance With Llm Reviewing Policy:**

Affirmed.

**Final Justification:**

I have increased my score from Weak Reject to Weak Accept due to the author's improved treatment of TIDE's computational overhead, limitations at high temperatures, and inclusion of other toxicity functions.

I am hesitant to raise my score further, as I feel the overhead and ineffectiveness at high temperatures are substantial weaknesses. However, these are now outweighed by the method's novelty and empirical effectiveness. It seems like a quality defense that offers a new perspective on toxicity mitigation.

**Key Questions For Authors:**

1. What is the wall-clock runtime of TIDE, and how does it compare to the baseline methods such as RAD [1]? More generally, a clearer discussion and comparison of TIDE’s computational complexity would potentially help address Weakness 1.

2. Does the method generalize to other toxicity detectors, e.g., Detoxify [2] or ToxiGen-RoBERTa [3]? Strong generalizability would help address Weakness 2 and enable a fairer empirical evaluation.

3. Does the ineffectiveness at higher temperatures apply to the other baselines, or is it a unique limitation of TIDE? Can higher temperatures be effective if we also increase the number of samples ($N$)? This would help clarify under what conditions certain temperatures can be used, potentially addressing Weakness 4.

[1] Deng and Raffel, “Reward-Augmented Decoding: Efficient Controlled Text Generation With a Unidirectional Reward Model,” EMNLP 2023.

[2] https://github.com/unitaryai/detoxify

[3] Hartvigsen et al., “ToxiGen: A Large-Scale Machine-Generated Dataset for Adversarial and Implicit Hate Speech Detection,” ACL 2022.

**Limitations:**

In the impact statement, the authors state, “We have not identified a direct or practical way to adapt the proposed method to reliably induce such malicious behaviors under these constraints.” However, to my understanding, couldn't we just optimize $1 - \Phi(X)$ (or perform gradient ascent on $\Phi(X)$) to generate high-toxicity outputs? Unless I am misunderstanding something, this type of simple misuse should be properly addressed in the impact statement.

**Strengths And Weaknesses:**

### **Strengths**

1. (Originality) To my knowledge, this is the first zero-order optimization method to address toxicity in LLMs. The approach seems novel and distinct from prior decoding-time approaches.

2. (Significance) TIDE is empirically effective, generally achieving greater toxicity reduction than the baselines with small fluency loss. Notably, it achieves a near-zero “Toxic Rate” across evaluation configurations, providing strong evidence that the zero-order optimization is functional.

3. (Significance) TIDE does not require any training data and is compatible with any (quantitative) toxicity evaluator, making it flexible to different deployment scenarios.

### **Weaknesses**

1. (Soundness/Significance) The discussion surrounding integration with vLLM seems overstated, particularly that batched inference can largely amortize the deployment costs of TIDE. For instance: (i) TIDE updates the prompt embeddings iteratively, so you cannot batch the multiple requests required per input at once, i.e., the single-input latency is still significantly increased; (ii) Every generation must be sent to the toxicity classifier, which would incur significant slowdown and is not readily integrable with vLLM; (iii) Even with batched inference, TIDE still increases the total number of requests linearly in $\bar{K}$, and thus would have a large impact on final response throughput; (iv) Updating the prompt embeddings would invalidate all KV cache states, vastly reducing the computational savings offered by shared KV caching. Overall, I think the computational burden of TIDE is significantly underestimated, making it a major weakness of the proposed approach.

2. (Soundness) The empirical evaluation seems a bit unfair, as TIDE is the only method with direct access to the “ground-truth” evaluation function (Perspective API) during generation. I think this is reasonable, as TIDE is also the only method that explicitly requires such a function, but it is nonetheless an implicit advantage not discussed in the paper. A possible way to remediate this would be to vary the classifier used for generation versus evaluation.

3. (Soundness) Toxicity is a high-variance behavior [1], making $M=3$ a concerningly small trial count (most prior works use $M=25$ [2, 3, 4]). Could the authors add error bars or some form of statistical significance to the results?

4. (Significance) The ineffectiveness of TIDE at higher temperatures makes the method brittle. Furthermore, to my knowledge, the standard temperature for toxicity evaluations is $T=1$ [1], so most prior work does not suffer from this limitation.

5. (Presentation) (Minor) There is a lot of white space in the main text. I think some parts in the Appendix (e.g., the sensitivity analysis) could be moved to the main text to improve the presentation quality.

[1] Gehman et al., “RealToxicityPrompts: Evaluating Neural Toxic Degeneration in Language Models,” EMNLP 2020.

[2] Deng and Raffel, “Reward-Augmented Decoding: Efficient Controlled Text Generation With a Unidirectional Reward Model,” EMNLP 2023.

[3] Ko et al., “Large Language Models can Become Strong Self-Detoxifiers,” ICLR 2025.

[4] Liu et al., “DExperts: Decoding-Time Controlled Text Generation with Experts and Anti-Experts,” ACL 2021.

---

> ### Author Rebuttal · Authors · 2026-03-30
>
> We thank the reviewer for their constructive evaluation. We address each point below. ("Q" = Question, "W" = Weakness)
>
> _Due to the character limit, some details are kept brief; we are happy to elaborate in a follow-up response upon request._
>
> ---
>
> **Q1.** We present wall-clock runtimes of TIDE vs. RAD ($\beta=0.75$) per prompt on the RTP challenging subset.
>
> | Model | TIDE Mean | TIDE Median | RAD Mean
> |---|:---:|:---:|:---:|
> | Llama 3.1-8B | 8.99 s | 3.25 s | 1.00 s |
> | Qwen3-4B | 2.93 s | 1.96 s | 1.16 s |
> | Gemma 2-2B | 4.54 s | 2.25 s | 1.17 s |
> | GPT-2 Large | 4.42 s | 1.63 s | 0.61 s |
>
> TIDE is slower per input due to iterative optimization. Given that TIDE requires no preparation in advance (e.g., training models or additional data), we believe this cost becomes moderate especially when compared to the toxicity improvements it brings. Further, the median is substantially lower than the mean: early stopping resolves most prompts quickly, with the mean inflated by a few hard cases.
>
> **Regarding W1:** We agree the vLLM discussion overstates how much batching remedies the iterative cost—the intended scope is amortizing the $N$ forward passes *per iteration*, not eliminating iterative overhead. We will clarify this.
>
> ---
>
> **Q2.** TIDE is agnostic to the toxicity function—any continuous, real-valued scorer is compatible without modification. To verify, we re-evaluated all methods' already-collected outputs using Detoxify (no additional optimization). Max-toxicity under Detoxify (RTP-challenging):
>
> | Model | TIDE | RAD* |
> |---|:-----:|:-----|
> | Llama 3.1-8B | 0.123 | 0.127 ($\beta$=75) |
> | Qwen3-4B | 0.091 | 0.237 ($\beta$=10) |
> | Gemma 2-2B | 0.112 | 0.145 ($\beta$=50), 0.103 ($\beta$=75) |
> | GPT-2 Large | 0.114 | 0.121 ($\beta$=50) |
>
> RAD* = configuration(s) at perplexity closest to TIDE's (per the paper); for Gemma, TIDE's perplexity falls between the two entries. At comparable perplexity, TIDE again achieves lower toxicity; higher-$\beta$ RAD configurations that surpass TIDE come with substantially higher perplexity. Base-model toxicity under Detoxify is comparable to Perspective API, confirming this consistency is not an artifact of a more lenient evaluator. Overall, TIDE can generalize also when the optimization and inference evaluators are different.
>
> **Regarding W2:** We agree this asymmetry should be discussed and note, however, that it is bidirectional — RAD requires a trained reward model; SASA learns linear parameters. TIDE essentially replaces these requirements with direct queries to the oracle function. We will discuss this to Section 6.1.
>
> ---
>
> **Q3.** Our framing in Section 6.1 was imprecise: the zeroth-order estimator extends to stochastic objectives. Therefore, _TIDE can be applied at high temperatures._ Low temperature is chosen for efficiency—at higher $T$, larger $N$ is required to compensate for increased generation stochasticity and thus maintain reliable gradient estimates, raising cost.
>
> $T=1.0$ in prior work *stress-tests decoding-based methods* by manipulating the sampling distribution; TIDE does not modify decoding, so this framing does not apply. The temperature sensitivity is inherent to TIDE's zeroth-order mechanism, whereas decoding-based methods are designed to be robust to sampling noise, though may be sensitive in other ways (e.g., SASA is ineffective at low temperature). Section 6.1 will be clarified in the revision.
>
> At $T=0.1$, generations across trials are already very close. $M=25$ is necessary at $T=1.0$ because sampling variance is high; at $T=0.1$, the mean behavior is already captured by $M=3$. For instance, three Qwen3-4B completions for the same RTP prompt:
>
> ```
> "...You, My Love, My Life\n\nOkay, so I need to figure out what the user is",
> "...You, My Love\n\nOkay, so I need to write a diary entry for the Secret Diary of",
> "...You, My Love\n\nOkay, so I need to write a diary entry for the Secret Diary of"
> ```
>
> We further conducted Wilcoxon signed-rank tests (one-sided) on per-prompt max-toxicity across the 1199 RTP prompts:
>
> - **TIDE vs. Base model:** Statistically significant on all four models ($p \approx 0$, win rate 72–86%).
> - **TIDE vs. RAD ($\beta=75$)**: Per-prompt max-toxicity is statistically comparable, expected as RAD is the closest baseline. However, TIDE achieves dramatically lower toxic rates (0.2–6.0% vs. 6.6–9.8%) and lower perplexity.
>
> Separate tests on toxic rates confirm significant differences ($p \approx 0$) for both comparisons. The comparable max-toxicity combined with significantly lower toxic rate shows TIDE is particularly effective at eliminating the worst outputs.
>
> ---
>
> **W5.** This is mainly caused by the LaTeX template and we agree: we will move the sensitivity analysis to the main text by using the additional page if the paper is accepted.
>
> ---
>
> **Limitations (misuse).** This is a valid concern that we overlooked in the draft. We thank the reviewer for raising it and will revise the Impact Statement accordingly.

---

> > ### Author Rebuttal · Reviewer_QvLT · 2026-04-03
> >
> > I thank the authors for their responses. I think most of my concerns have been adequately addressed; however, I am still hesitant about the method's core efficiency and its compatibility with higher temperatures.
> >
> > For efficiency, while it is no doubt an advantage that TIDE is training-free, the massive increase in compute required by multiple forward passes is still not moderate; over time, this compute will likely exceed that incurred by a training-time method like DPO. RAD is already a relatively inefficient method, so the fact that TIDE's average latency is ~2–8$\times$ higher is a large limitation.
> >
> > For the temperature, I don't think it is valid to characterize $T=1$ as a stress-testing scenario. To my knowledge, temperatures in the range of 0.7–1.0 are typical for standard modern deployments, making the usage of 0.1 impractical.
> >
> > Nevertheless, as the response at least partially addresses these aspects, and the method brings a novel technique to the toxicity mitigation literature, I believe the strengths of this work now outweigh the limitations, and I will raise my score accordingly. I encourage the authors to clearly situate these limitations in the revision.

---

> > > ### Author Response · Authors · 2026-04-03
> > >
> > > Thank you very much for the constructive follow-up and for the score increase.
> > >
> > > As suggested, we will clearly situate both the efficiency overhead and the temperature sensitivity as limitations in the revised paper.
> > >
> > > For completeness of the rebuttal, we share results from high-temperature experiments initiated during the rebuttal period. To reduce the variance of the zeroth-order estimator at high $T$, we average $\Phi$ over $M' = 8$ independent completions per evaluation, with $N = 8$ Monte Carlo samples. Results on RTP-challenging at $T = 1.0$ with $M = 25$ trials, GPT-2 Large (baselines reproduced using the codebase of [1]):
> > >
> > > | Method | Avg Max Toxicity | Toxic Rate | Perplexity |
> > > |:---|:---:|:---:|:---:|
> > > | Base model | 0.912 | 0.99 | 7.66 |
> > > | RAD ($\beta = 10$) | 0.867 | 0.96 | 8.01 |
> > > | RAD ($\beta = 50$) | 0.519 | 0.48 | 10.79 |
> > > | RAD ($\beta = 75$) | 0.40 | 0.33 | 11.30 |
> > > | RAD ($\beta = 100$) | 0.366 | 0.22 | 12.22 |
> > > | SASA ($\beta = 10$) | 0.837 | 0.95 | 8.39 |
> > > | SASA ($\beta = 50$) | 0.501 | 0.57 | 11.53 |
> > > | SASA ($\beta = 75$) | 0.451 | 0.47 | 12.89 |
> > > | SASA ($\beta = 100$) | 0.448 | 0.44 | 13.57 |
> > > | TIDE ($M' = 8$, $N = 8$) | **0.32** | **0.0** | 13.98 |
> > >
> > > **At $T = 1.0$, TIDE still achieves the lowest maximum toxicity and zero toxic rate—while operating entirely at the input level.** The average iteration count also drops to $\bar{K} = 2.22$ from $3.17$ at low temperature as the smoothed objective reduces the variance of the zeroth-order gradient estimator.
> > >
> > > Results for Qwen3-4B are in progress and all high-temperature results will be included in the revision.
> > >
> > > We appreciate the discussion; it has helped strengthen the paper.
> > >
> > > ---
> > >
> > > ##### [1] Ko et al., _"Large Language Models can be Strong Self-Detoxifiers,"_ ICLR 2024.

---

### Official Review · Reviewer_Tv7N · 2026-03-03

**Soundness:** 2
**Presentation:** 4
**Significance:** 3
**Originality:** 4
**Overall Recommendation:** 5
**Confidence:** 3

**Summary:**

The paper introduces a novel zeroth-order optimization (ZO) test-time detoxification approach that only needs access to input embeddings, a toxicity score function and forward evaluations, without requiring access to the full model. The main idea is to treat the prompt's input embeddings as continuous control variables. By using ZO the method iteratively updates the input embeddings to reduce output's toxicity.

**Compliance With Llm Reviewing Policy:**

Affirmed.

**Final Justification:**

The rebuttal addressed my main concerns and I increase my evaluation (4>5).

**Key Questions For Authors:**

To assess generation quality, you measure fluency perplexity using a larger model from the same family. Why not use the same model over all families? That would lead to more consistency in the evaluation.

You focus on plug-and-play methods that do not require retraining the model or an auxiliary module. Why is that? I would like to see a comparison of less memory efficient methods compare in wall-clock time and toxicity improvements.

You use a toxicity threshold of k=0.5. Why limit it to that? An ablation on this would be interesting to see how much more toxicity can be reduced with your method. Moreover, I would like to see the wall-clock time at different K (number of iterations).

In the paper authors write “across all prompts and trials, the decoded optimized embeddings are identical to the original prompts”. Doesn’t that mean that there were no significant changes to embedding?  Why would that have any effect on the LLM but not in the decoder? (This seems pretty odd to me).  Can you provide a detailed analysis and a quantitative evaluation on this?

**Limitations:**

Yes

**Strengths And Weaknesses:**

**Strengths**

The method does not need access to external models, no fine-tuning, and no access to full model-weights. They only require an external toxicity measurement tool, which is standard in this kind of methods.

ZO is well grounded in optimization literature, and the paper adapts it effectively for the LLM use case.

The results show that not many iterations are needed in order to achieve the toxicity score desired of 0.5.

By not required gradient calculations over the whole model it can be more memory efficient than related methods which fine-tune the models or require gradient calculations over the full model.

**Weaknesses**

The method requires multiple forward passes per generation, adding computation overhead. This could be a real limitation to real applications. Since the mode is not being retrained, that means it needs to be done every time you query the model.

The method seems to be ineffective on medium to high temperature values.

The evaluation is limited to plug-and-play methods that do not require retraining the model or an auxiliary module.

For a method that could some significant overhead, wall-clock time comparisons are missing.

---

> ### Author Rebuttal · Authors · 2026-03-30
>
> Thank you for your comprehensive evaluation. We address each point below. ("Q" = Question, "W" = Weakness).
>
> _Due to the character limit, some details are kept brief; we are happy to elaborate in a follow-up response upon request._
>
> ---
>
> **Q1.** A global evaluator is feasible but introduces systematic cross-family bias. Model families can differ in pretraining data and tokenization, so a sequence natural to one family may be unlikely under another—inflating perplexity independently of actual fluency. To verify, we evaluated base-model outputs on RTP-challenging using Qwen3-14B (larger than every evaluated model) as a single global evaluator:
>
> | Model | Same-family PPL | Global PPL (Qwen3-14B) |
> |---|:---:|:---:|
> | GPT-2 Large | 4.13 | 5.07 |
> | Gemma 2-2B | 3.36 | 5.31 |
> | Llama 3.1-8B | 3.54 | 4.46 |
> | Qwen3-4B | 4.28$^*$ | 4.09 |
>
> $(^*)$ = measured by Qwen3-8B.
>
> Perplexity increases by 23–58\% for non-Qwen models, while remaining largely unchanged for Qwen. This bias can invert expected orderings: under a single Qwen evaluator, the larger model (Llama) would thus appear less fluent—even though same-family evaluation confirms Llama is indeed more fluent (3.54 vs. 4.28). Same-family evaluation avoids this by measuring fluency relative to each model's own distribution. We will clarify this rationale in the paper.
>
> ---
>
> **W2.** Our framing in Section 6.1 was not sufficiently clear on this point. _TIDE can operate at higher temperatures_—doing so simply requires more Monte Carlo samples ($N$) to compensate for amplified generation noise, which reduces test-time efficiency. We will correct Section 6.1 to reflect that the low-temperature regime is a practical efficiency choice rather than a theoretical constraint.
>
> ---
>
> **Q2.** *(Also addresses W1, W3, W4.)* Our scope is detoxifying pretrained models as released; retraining changes the evaluation object, making comparison with TIDE's training-free setting no longer apples-to-apples. We do, however, include RAD and SASA which train auxiliary modules; RAD trains a GPT-2 reward model, SASA learns linear parameters prior to training.
>
> Following the reviewer's suggestion, we report per-prompt wall-clock runtimes of TIDE vs. RAD ($\beta=75$; configuration closest to ours) on RTP-challenging.
>
> | Model | TIDE Mean | TIDE Median | RAD Mean | $\bar{K}$ |
> |---|:------:|:------:|:------:|----: |
> | Llama 3.1-8B | 8.99 s | 3.25 s | 1.00 s | 3.15 |
> | Qwen3-4B | 2.93 s | 1.96 s | 1.16 s | 1.63 |
> | Gemma 2-2B | 4.54 s | 2.25 s | 1.17 s | 2.40 |
> | GPT-2 Large | 4.42 s | 1.63 s | 0.61 s | 3.17 |
>
> TIDE is slower per input due to iterative optimization. Given that TIDE requires no preparation in advance (e.g., training models or additional data), we believe this cost becomes moderate especially when compared to the toxicity improvements it brings. Further, the median is substantially lower than the mean: early stopping resolves most prompts quickly, with the mean inflated by a few high-toxicity inputs needing more iterations (see $\bar{K}$ in the table). On less challenging benchmarks (AttaQ and BOLD), $\bar{K}$ already drops to approximately 1.
>
> Additionally, Wilcoxon signed-rank tests (one-sided) on per-prompt toxic rates confirm that TIDE's reductions over the base model and RAD are statistically significant ($p \approx 0$): TIDE at 0.2–6.0% vs. RAD's 6.6–9.8%.
>
> ---
>
> **Q3.** We follow prior work [1] in setting $\tau=0.5$. Perspective API calibrates its toxicity scores via isotonic regression to output the probability that text is toxic (88% agreement with human raters [1]), so $\tau=0.5$ means "more likely toxic than not." Lowering $\tau$ further would yield additional reduction at the cost of more iterations and potential fluency degradation; $\tau=0.5$ balances effectiveness and compute.
>
> **Wall-clock time at different $\bar{K}$:** Please see our response to Q2.
>
> All of these results will be added to the revised paper.
>
> ---
>
> **Q4.** Identical decoding does not mean insignificant changes; rather, it indicates that the perturbations are sub-lexical. Our optimization moves embeddings within their original token regions (hence identical decoding) but the model receives different continuous inputs. This continuous shift propagates through the transformer's smooth operations and effectively alters the output distribution.
>
> The toxicity reduction itself is quantitative evidence: decoded prompts remain identical, yet max-toxicity drops substantially (e.g., $0.591 \to 0.156$ for GPT-2 on RTP). This confirms that generation is sensitive to continuous embedding variations within the same token subspace. This observation is a broader finding of independent interest of our study.
>
> We will add this mechanistic interpretation along with per-token embedding distance statistics to the revision. We also welcome any specific quantitative analyses the reviewer may have in mind.
>
> ---
>
> #####
>
> [1] Gehman et al., "RealToxicityPrompts: Evaluating Neural Toxic Degeneration in Language Models."

---

> > ### Author Rebuttal · Reviewer_Tv7N · 2026-03-31
> >
> > I would like to thank the authors for their responses. I have decided to update my score.

---

> > > ### Author Response · Authors · 2026-04-02
> > >
> > > Thank you for carefully considering our responses and for updating your assessment. We appreciate the constructive exchange.

---

### Decision · Program_Chairs · 2026-04-30

**Decision:**

Accept (regular)

**Comment:**

This work proposes a training-free, test-time detoxification approach that operates in fully black-box settings to mitigate toxic text generation in large language models. The method approximates gradients of toxicity with respect to input embeddings via zeroth-order optimization, enabling efficient steering of model outputs using only forward passes and a toxicity scorer. Empirical results demonstrate consistent and robust reductions in toxicity across models and prompts while maintaining strong generation quality. During the rebuttal, the authors successfully addressed the reviewers' concerns, and all reviewers agreed to accept this work. We strongly recommend that the authors incorporate the reviewers’ feedback and the additional results presented in the rebuttal into the final version.

Congratulations!